# Asymptotic Analysis of the *k*th Subword Complexity

**DOI:** 10.3390/e22020207

**Published:** 2020-02-12

**Authors:** Lida Ahmadi, Mark Daniel Ward

**Affiliations:** 1Department of Mathematics, Purdue University, West Lafayette, IN 47907, USA; 2Department of Statistics, Purdue University, West Lafayette, IN 47907, USA; mdw@purdue.edu

**Keywords:** subword complexity, asymptotics, generating functions, saddle point method, probability, the Mellin transform, moments

## Abstract

Patterns within strings enable us to extract vital information regarding a string’s randomness. Understanding whether a string is random (Showing no to little repetition in patterns) or periodic (showing repetitions in patterns) are described by a value that is called the *k*th Subword Complexity of the character string. By definition, the *k*th Subword Complexity is the number of distinct substrings of length *k* that appear in a given string. In this paper, we evaluate the expected value and the second factorial moment (followed by a corollary on the second moment) of the *k*th Subword Complexity for the binary strings over memory-less sources. We first take a combinatorial approach to derive a probability generating function for the number of occurrences of patterns in strings of finite length. This enables us to have an exact expression for the two moments in terms of patterns’ auto-correlation and correlation polynomials. We then investigate the asymptotic behavior for values of k=Θ(logn). In the proof, we compare the distribution of the *k*th Subword Complexity of binary strings to the distribution of distinct prefixes of independent strings stored in a trie. The methodology that we use involves complex analysis, analytical poissonization and depoissonization, the Mellin transform, and saddle point analysis.

## 1. Introduction

Analyzing and understanding occurrences of patterns in a character string is helpful for extracting useful information regarding the nature of a string. We classify strings to low-complexity and high-complexity, according to their level of randomness. For instance, we take the binary string X=10101010..., which is constructed by repetitions of the pattern w=10. This string is periodic, and therefore has low randomness. Such periodic strings are classified as low-complexity strings, whereas strings that do not show periodicity are considered to have high complexity. An effective way of measuring a string’s randomness is to count all distinct patterns that appear as contiguous subwords in the string. This value is called the Subword Complexity. The name is given by Ehrenfeucht, Lee, and Rozenberg [1], and initially was introduced by Morse and Hedlund in 1938 [2]. The higher the Subword Complexity, the more complex the string is considered to be.

Assessing information about the distribution of the Subword Complexity enables us to better characterize strings, and determine atypically random or periodic strings that have complexities far from the average complexity [3]. This type of string classification has applications in fields such as data compression [4], genome analysis (see [5,6,7,8,9]), and plagiarism detection [10]. For example, in data compression, a data set is considered compressible if it has low complexity, as consists of repeated subwords. In computational genomics, Subword Complexity (known as k-mers) is used in detection of repeated sequences and DNA barcoding [11,12]. *k*-mers are composed of A, T, G, and C nucleotides. For instance, 7-mers for a DNA sequence GTAGAGCTGT is four, meaning that there are 4-hour distinct substrings of length 7 in the given DNA sequence. Counting *k*-mers becomes challenging for longer DNA sequences. Our results can be easily extended to the alphabet {A,T,G,C} and directly applied in theoretical analysis of the genomic *k*-mer distributions under the Bernoulli probabilistic model, particularly when the length *n* of the sequence approaches infinity.

There are two variations for the definition of the Subword Complexity: the one that counts all distinct subwords of a given string (also known as Complexity Index and Sequence Complexity [13]), and the one that only counts the subwords of the same length, say *k*, that appear in the string. In our work, we analyze the latter, and we call it the *k*th Subword Complexity to avoid any confusion.

Throughout this work, we consider the *k*th Subword Complexity of a random binary string of length *n* over a memory-less source, and we denote it by $X_{n,k}$. We analyze the first and second factorial moments of $X_{n,k}$ (1) for the range k=Θ(logn), as n→∞. More precisely, will divide the analysis into three ranges as follows.

i.$\frac{1}{\log q^{-1}}\log n<k<\frac{2}{\log q^{-1}+\log p^{-1}}\log n$,ii.$\frac{2}{\log q^{-1}+\log p^{-1}}\log n<k<\frac{1}{q\log q^{-1}+p\log p^{-1}}\log n$, andiii.$\frac{1}{q\log q^{-1}+p\log p^{-1}}\log n<k<\frac{1}{\log p^{-1}}\log n$.

Our approach involves two major steps. First, we choose a suitable model for the asymptotic analysis, and afterwards we provide proofs for the derivation of the asymptotic expansion of the first two factorial moments.

### 1.1. Part I

This part of the analysis is inspired by the earlier work of Jacquet and Szpankowski [14] on the analysis of suffix trees by comparing them to independent tries. A trie, first introduced by René de la Briandais in 1959 (see [15]), is a search tree that stores *n* strings, according to their prefixes. A suffix tree, introduced by Weiner in 1973 (see [16]), is a trie where the strings are suffixes of a given string. An example of these data structures are given in Figure 1.

A direct asymptotic analysis of the moments is a difficult task, as patterns in a string are not independent from each other. However, we note that each pattern in a string can be regarded as a prefix of a suffix of the string. Therefore, the number of distinct patterns of length *k* in a string is actually the number of nodes of the suffix tree at level *k* and lower. It is shown by I. Gheorghiciuc and M. D. Ward [17] that the expected value of the *k*-th Subword Complexity of a Bernoulli string of length *n* is asymptotically comparable to the expected value of the number of nodes at level *k* of a trie built over *n* independent strings generated by a memory-less source.

We extend this analysis to the desired range for *k*, and we prove that the result holds for when *k* grows logarithmically with *n*. Additionally, we show that asymptotically, the second factorial moment of the *k*-th Subword Complexity can also be estimated by admitting the same independent model generated by a memory-less source. The proof of this theorem heavily relies on the characterization of the overlaps of the patterns with themselves and with one another. Autocorrelation and correlation polynomials explicitly describe these overlaps. The analytic properties of these polynomials are key to understanding repetitions of patterns in large Bernoulli strings. This, in conjunction with Cauchy’s integral formula (used to compare the generating functions in the two models) and the residue theorem, provides solid verification that the second factorial moment in the Subword Complexity behaves the same as in the independent model.

To make this comparison, we derive the generating functions of the first two factorial moments in both settings. In a paper published by F. Bassino, J. Clément, and P. Nicodème in 2012 [18], the authors provide a multivariate probability generating function f(z,x) for the number of occurrences of patterns in a finite Bernoulli string. That is, given a pattern *w*, the coefficient of the term $z^{n}x^{m}$ in f(z,x) is the probability in the Bernoulli model that a random string of size *n* has exactly *m* occurrences of the pattern *w*. Following their technique, we derive the exact expression for the generating functions of the first two factorial moments of the *k*th Subword Complexity. In the independent model, the generating functions are obtained by basic probability concepts.

### 1.2. Part II

This part of the proof is analogous to the analysis of profile of tries [19]. To capture the asymptotic behavior, the expressions for the first two factorial moments in the independent trie are further improved by means of a Poisson process. The poissonized version yields generating functions in the form of harmonic sums for each of the moments. The Mellin transform and the inverse Mellin transforms of these harmonic sums establish a connection between the asymptotic expansion and singularities of the transformed function. This methodology is sufficient for when the length *k* of the patterns are fixed. However, allowing *k* to grow with *n*, makes the analysis more challenging. This is because for large *k*, the dominant term of the poissonized generating function may come from the term involving *k*, and singularities may not be significant compared to the growth of *k*. This issue is treated by combining the singularity analysis with a saddle point method [20]. The outcome of the analysis is a precise first-order asymptotics of the moments in the poissonized model. Depoissonization theorems are then applied to obtain the desired result in the Bernoulli model.

## 2. Results

For a binary string $X=X_{1}X_{2}...X_{n}$, where $X_{i}$’s (i=1,...,n) are independent and identically distributed random variables, we assume that $P(X_{i}=1)=p$, $P(X_{i}=0)=q=1-p$, and p>q. We define the *k*th Subword Complexity, $X_{n,k}$, to be the number of distinct substrings of length *k* that appear in a random string *X* with the above assumptions. In this work, we obtain the first order asymptotics for the average and the second factorial moment of $X_{n,k}$. The analysis is done in the range k=Θ(logn). We rewrite this range as k=alogn, and by performing a saddle point analysis, we will show that
(1)$$\begin{array}{c} 1/\log q^{-1}<a<1/\log p^{-1} \end{array}$$

In the first step, we compare the *k*th Subword Complexity to an independent model constructed in the following way: We store a set of *n* independently generated strings by a memory-less source in a trie. This means that each string is a sequence of independent and identically distributed Bernoulli random variables from the binary alphabet A={0,1}, with P(1)=p, P(0)=q=1−p. We denote the number of distinct prefixes of length *k* in the trie by ${\hat{X}}_{n,k}$, and we call it *the kth prefix complexity*. Before proceeding any further, we remind that factorial moments of a random variable are defined as following.

**Definition** **1.**
*The jth factorial moment of a random variable X is defined as*
(2)$$\begin{array}{c} E\left[{\left(X\right)}_{j}\right]=E\left[\left(X\right)\left(X-1\right)\left(X-2\right)...\left(X-j+1\right)\right], \end{array}$$
*where j = 1, 2, … will show that the first and second factorial moments of $X_{n,k}$ are asymptotically comparable to those of ${\hat{X}}_{n,k}$, when k=Θ(logn). We have the following theorems.*


**Theorem** **1.**
*For large values of n, and for k=Θ(logn), there exists M>0 such that*
$$E\left[X_{n,k}\right]-E\left[{\hat{X}}_{n,k}\right]=O\left(n^{-M}\right).$$


We also prove a similar result for the second factorial moments of the *k*th Subword Complexity and the *k*th Prefix Complexity:

**Theorem** **2.**
*For large values of n, and for k=Θ(logn), there exists ϵ>0 such that*
$$E\left[{\left(X_{n,k}\right)}_{2}\right]-E\left[{\left({\hat{X}}_{n,k}\right)}_{2}\right]=O\left(n^{-\epsilon }\right).$$


In the second part of our analysis, we derive the first order asymptotics of the *k*th Prefix Complexity. The methodology used here is analogous to the analysis of profile of tries [19]. The rate of the asymptotic growth depends on the location of the value *a* as seen in (Equation 1). For instance, for the average *k*th Subword Complexity, $E[X_{n,k}]$, we have the following observations.

i.For the range $I_{1}:\frac{1}{\log q^{-1}}<a<\frac{2}{\log q^{-1}+\log p^{-1}}$, the growth rate is of order $O(2^{k})$,ii.in the range $I_{2}:\frac{2}{\log q^{-1}+\log p^{-1}}<a<\frac{1}{q\log q^{-1}+p\log p^{-1}}$, we observe some oscillations with *n*, andiii.in the range $I_{3}:\frac{1}{q\log q^{-1}+p\log p^{-1}}<a<\frac{1}{\log p^{-1}}$, the average has a linear growth O(n).

The above observations will be discussed in depth in the proofs of the following theorems.

**Theorem** **3.**
*The average of the kth Prefix Complexity has the following asymptotic expansion*
*i.* 
*For $a\in I_{1}$,*
(3)$$\begin{array}{cc} E[{\hat{X}}_{n,k}] & =2^{k}-\Phi_{1}\left(\left(1+\log p\right)\log_{p/q}n\right)\frac{n^{\nu }}{\sqrt{\log n}}\;\left(1+O\left(\frac{1}{\sqrt{\log n}}\right)\right), \end{array}$$

*where $\nu =-r_{0}+a\log\left(p^{-r_{0}}+q^{-r_{0}}\right)$, and*
$$\begin{array}{cc} \Phi_{1}\left(x\right) & =\frac{{\left(p/q\right)}^{-r_{0}/2}+{\left(p/q\right)}^{r_{0}/2}}{\sqrt[{}]{2\pi }\log p/q}\sum_{j\in Z}\Gamma \left(r_{0}+it_{j}\right)e^{-2\pi ijx} \end{array}$$

*is a bounded periodic function.*
*ii.* 
*For $a\in I_{2}$,*
$$\begin{array}{cc} E[{\hat{X}}_{n,k}] & =\Phi_{1}\left(\left(1+\log p\right)\log_{p/q}n\right)\frac{n^{\nu }}{\sqrt{\log n}}\;\left(1+O\left(\frac{1}{\sqrt{\log n}}\right)\right). \end{array}$$
*iii.* 
*For $a\in I_{3}$*
$$\begin{array}{cc} E[{\hat{X}}_{n,k}] & =n+O(n^{\nu_{0}}), \end{array}$$

*for some $\nu_{0}<1$.*



**Theorem** **4.**
*The second factorial moment of the kth Prefix Complexity has the following asymptotic expansion.*
*i.* 
*For $a\in I_{1}$,*
$$\begin{array}{c} E[{\left({\hat{X}}_{n,k}\right)}_{2}]={\left(2^{k}-\Phi_{1}\left(\log_{p/q}n\left(1+\log p\right)\right)\frac{n^{\nu }}{\sqrt{\log n}}\left(1+O\left(\frac{1}{\sqrt{\log n}}\right)\right)\right)}^{2}. \end{array}$$
*ii.* 
*For $a\in I_{2}$,*
$$\begin{array}{cc} E[{\left({\hat{X}}_{n,k}\right)}_{2}] & =\Phi_{1}^{2}\left(\log_{p/q}n\left(1+\log p\right)\right)\frac{n^{2\nu }}{\log n}\left(1+O\left(\frac{1}{\log n}\right)\right). \end{array}$$
*iii.* 
*For $a\in I_{3}$,*
$$\begin{array}{c} E\left[{\left({\hat{X}}_{n,k}\right)}_{2}\right]=n^{2}+O\left(n^{2\nu_{0}}\right). \end{array}$$



The periodic function $\Phi_{1}\left(x\right)$ in Theorems 3 and 4 is shown in Figure 2.

The results in Theorem 4 will follow for the second moment of the *k*th Subword Complexity as the analysis can be easily extended from the second factorial moment to the second moment. The variance however, as seen in Figure 3, does not show the same asymptotic behavior as the variance of *k*th Subword Complexity.

## 3. Proofs and Methods

### 3.1. Groundwork

We first introduce a few terminologies and lemmas regarding overlaps of patterns and their number of occurrences in texts. Some of the notations we use in this work are borrowed from [18] and [21].

**Definition** **2.**
*For a binary word $w=w_{1}...w_{k}$ of length k, The autocorrelation set $S_{w}$ of the word w is defined in the following way.*
(4)$$\begin{array}{c} S_{w}=\left\{w_{i+1}...w_{k}\;|\;w_{1}...w_{i}=w_{k-i+1}...w_{k}\right\}. \end{array}$$

*The autocorrelation index set is*
(5)$$\begin{array}{c} P\left(w\right)=\left\{i\;|\;w_{1}...w_{i}=w_{k-i+1}...w_{k}\right\}, \end{array}$$

*And the autocorrelation polynomial is*
(6)$$\begin{array}{c} S_{w}\left(z\right)=\sum_{i\in P(w)}P\left(w_{i+1}...w_{k}\right)z^{k-i}. \end{array}$$


**Definition** **3.**
*For the distinct binary words $w=w_{1}...w_{k}$ and $w^{\prime }=w_{1}^{\prime }...w_{k}^{\prime }$, the correlation set $S_{w,w^{\prime }}$ of the words w and $w^{\prime }$ is*
(7)$$\begin{array}{c} S_{w,w^{\prime }}=\left\{w_{i+1}^{\prime }...w_{k}^{\prime }\;|\;w_{1}^{\prime }...w_{i}^{\prime }=w_{k-i+1}...w_{k}\right\}. \end{array}$$
*The correlation index set is*
(8)$$\begin{array}{c} P\left(w,w^{\prime }\right)=\left\{i\;|\;w_{1}^{\prime }...w_{i}^{\prime }=w_{k-i+1}...w_{k}\right\}, \end{array}$$
*The correlation polynomial is*
(9)$$\begin{array}{c} S_{w,w^{\prime }}\left(z\right)=\sum_{i\in P(w,w^{\prime })}P\left(w_{i+1}^{\prime }...w_{k}^{\prime }\right)z^{k-i}. \end{array}$$


The following two lemmas present the probability generating functions for the number of occurrences of a single pattern and a pair of distinct pattern, respectively, in a random text of length *n*. For a detailed dissection on obtaining such generating functions, refer to [18].

**Lemma** **1.**
*The Occurrence probability generating function for a single pattern w in a binary text over a memoryless source is given by $F_{w}\left(z,x-1\right)$, where*
(10)$$\begin{array}{c} F_{w}\left(z,t\right)=\frac{1}{1-A\left(z\right)-\frac{tP\left(w\right)z^{k}}{1-t(S_{w}\left(z\right)-1)}}, \end{array}$$

*The coefficient $\left[z^{n}x^{m}\right]F_{w}\left(z,x-1\right)$ is the probability that a random binary string of length n has m occurrences of the pattern w.*


**Lemma** **2.**
*The Occurrence PGF for two distinct Patterns of length k in a Bernoulli random text is given by $F_{w,w^{\prime }}\left(z,x_{1}-1,x_{2}-1\right)$ where,*
(11)$$\begin{array}{c} F_{w,w^{\prime }}\left(z,t_{1},t_{2}\right)=\frac{1}{1-A\left(z\right)-M\left(z,t_{1},t_{2}\right)}, \end{array}$$

*and*
$$M\left(z,t_{1},t_{2}\right)=\left(\begin{array}{cc} P\left(w\right)z^{k}t_{1} & P\left(w^{\prime }\right)z^{k}t_{2} \end{array}\right){\left(\begin{array}{c} I-\left(\begin{array}{cc} \left(S_{w}\left(z\right)-1\right)t_{1} & S_{w,w^{\prime }}\left(z\right)t_{2}\\ S_{w^{\prime },w}\left(z\right)t_{1} & \left(S_{w^{\prime }}\left(z\right)-1\right)t_{2} \end{array}\right) \end{array}\right)}^{-1}\left(\begin{array}{c} 1\\ 1 \end{array}\right).$$

*The coefficient $\left[z^{n}x_{1}^{m_{1}}x_{2}^{m_{2}}\right]F_{w,w^{\prime }}\left(z,x_{1}-1,x_{2}-1\right)$ is the probability that there are $m_{1}$ occurrences of w and $m_{2}$ occurrences of $w^{\prime }$ in a random string of length n.*


The above results will be used to find the generating functions for the first two factorial moments of the *k*th Subword Complexity in the following section.

### 3.2. Derivation of Generating Functions

**Lemma** **3.**
*For generating functions $H_{k}\left(z\right)=\sum_{n\geq 0}E\left[X_{n,k}\right]z^{n}$ and $G_{k}\left(z\right)=\sum_{n\geq 0}E\left[{\left(X_{n,k}\right)}_{2}\right]z^{n}$, we have*
*i.* 
(12)$$\begin{array}{c} H_{k}\left(z\right)=\sum_{w\in A^{k}}\left(\frac{1}{1-z}-\frac{S_{w}\left(z\right)}{D_{w}\left(z\right)}\right), \end{array}$$
*where $D_{w}\left(z\right)=P\left(w\right)z^{k}+\left(1-z\right)S_{w}\left(z\right)$, and *
*ii.* 
(13)$$\begin{array}{c} G_{k}\left(z\right)=\sum_{\begin{array}{c} w,w^{{}^{\prime }}\in A^{k}\\ w\neq w^{\prime } \end{array}}\left(\frac{1}{1-z}-\frac{S_{w}\left(z\right)}{D_{w}\left(z\right)}-\frac{S_{w^{{}^{\prime }}}\left(z\right)}{D_{w^{\prime }}\left(z\right)}+\frac{S_{w}\left(z\right)S_{w^{{}^{\prime }}}\left(z\right)-S_{w,w^{\prime }}\left(z\right)S_{w^{{}^{\prime }},w}\left(z\right)}{D_{w,w^{\prime }}\left(z\right)}\right), \end{array}$$
*where*
(14)$$\begin{array}{c} D_{w,w^{\prime }}\left(z\right)=\left(1-z\right)\left(S_{w}\left(z\right)S_{w^{\prime }}\left(z\right)-S_{w,w^{\prime }}\left(z\right)S_{w^{\prime },w}\left(z\right)\right)\\ +z^{k}\left(P\left(w\right)\left(S_{w^{\prime }}\left(z\right)-S_{w,w^{\prime }}\left(z\right)\right)+P\left(w^{\prime }\right)\left(S_{w}\left(z\right)-S_{w^{\prime },w}\left(z\right)\right)\right). \end{array}$$



**Proof.** i. We define
$$X_{n,k}^{\left(w\right)}=\left\{\begin{array}{cc} 1 & \;\mathrm{if}\;w\;\mathrm{appears}\;\mathrm{at}\;\mathrm{least}\;\mathrm{once}\;\mathrm{in}\;\mathrm{string}\;X\\ 0 & \;\mathrm{otherwise}. \end{array}\right.$$This yields
(15)$$\begin{array}{cc} E[X_{n,k}^{\left(w\right)}] & =P(X_{n,k}^{\left(w\right)}=1)\\ \; & =1-P(X_{n,k}^{\left(w\right)}=0)\\ \; & =1-\left[z^{n}x^{0}\right]F_{w}\left(z,x\right). \end{array}$$We observe that $\left[z^{n}x^{0}\right]F_{w}\left(z,x\right)=\left[z^{n}\right]F_{w}\left(z,0\right)$. By defining $f_{w}\left(z\right)=F_{w}\left(z,0\right)$ and from (Equation 10), we obtain
(16)$$\begin{array}{cc} f_{w}\left(z\right) & =\frac{S_{w}\left(z\right)}{P\left(w\right)z^{k}+\left(1-z\right)S_{w}\left(z\right)}. \end{array}$$Having the above function, we derive the following result.
(17)$$\begin{array}{cc} H(z) & =\sum_{n\geq 0}E\left[X_{n,k}\right]z^{n}\\ \; & =\sum_{n\geq 0}\sum_{w\in A^{k}}\left(1-\left[z^{n}\right]f_{w}\left(z\right)\right)z^{n}\\ \; & =\sum_{w\in A^{k}}\left(\frac{1}{1-z}-f_{w}\left(z\right)\right)\\ \; & =\sum_{w\in A^{k}}\left(\frac{1}{1-z}-\frac{S_{w}\left(z\right)}{D_{w}\left(z\right)}\right). \end{array}$$ii. For this part, we first note that
(18)$$\begin{array}{cc} E[{\left(X_{n,k}\right)}_{2}] & =E\left[X_{n,k}^{2}\right]-E\left[X_{n,k}\right]\\ \; & =E\left[{\left(X_{n,k}^{\left(w\right)}+...+X_{n,k}^{\left(w^{\left(r\right)}\right)}\right)}^{2}\right]-E\left[X_{n,k}^{\left(w\right)}+...+X_{n,k}^{(w^{\left(r\right)})}\right]\\ \; & =\sum_{w\in A^{k}}E\left[{\left(X_{n,k}^{\left(w\right)}\right)}^{2}\right]+\sum_{\begin{array}{c} w,w^{\prime }\in A^{k}\\ w\neq w^{\prime } \end{array}}E\left[X_{n,k}^{\left(w\right)}X_{n,k}^{\left(w^{\prime }\right)}\right]-\sum_{w\in A^{k}}E\left[X_{n,k}^{\left(w\right)}\right]\\ \; & =\sum_{\begin{array}{c} w,w^{\prime }\in A^{k}\\ w\neq w^{\prime } \end{array}}E\left[X_{n,k}^{\left(w\right)}X_{n,k}^{\left(w^{\prime }\right)}\right]. \end{array}$$Due to properties of indicator random variables, we observe that the expected value of the second factorial moment has only one term:
(19)$$\begin{array}{c} E\left[{\left(X_{n,k}\right)}_{2}\right]=\sum_{\begin{array}{c} w,w^{\prime }\in A^{k}\\ w\neq w^{\prime } \end{array}}E\left[X_{n,k}^{\left(w\right)}X_{n,k}^{\left(w^{\prime }\right)}\right]. \end{array}$$We proceed by defining a second indicator variable as following.
$$X_{n,k}^{\left(w\right)}X_{n,k}^{\left(w^{\prime }\right)}=\left\{\begin{array}{cc} 1 & \;\mathrm{if}\;X_{n,k}^{\left(w\right)}=X_{n,k}^{\left(w^{\prime }\right)}=1\;\\ 0 & \;\mathrm{otherwise}. \end{array}\right.$$This gives
$$\begin{array}{cc} E[X_{n,k}^{\left(w\right)}X_{n,k}^{\left(w^{\prime }\right)}] & =P\left(X_{n,k}^{\left(w\right)}=1,X_{n,k}^{\left(w^{\prime }\right)}=1\right)\\ \; & =1-P\left(X_{n,k}^{\left(w\right)}=0\cup X_{n,k}^{\left(w^{\prime }\right)}=0\right)\\ \; & =1-P\left(X_{n,k}^{\left(w\right)}=0\right)-P\left(X_{n,k}^{\left(w^{\prime }\right)}=0\right)+P\left(X_{n,k}^{\left(w\right)}=0,X_{n,k}^{\left(w^{\prime }\right)}=0\right). \end{array}$$Finally, we are able to express $E[{\left(X_{n,k}\right)}_{2}]$ in the following
(20)$$\begin{array}{cc} E[{\left(X_{n,k}\right)}_{2}] & =\sum_{\begin{array}{c} w,w^{\prime }\in A^{k}\\ w\neq w^{\prime } \end{array}}\left(1-\left[z^{n}\right]f_{w}\left(z\right)-\left[z^{n}\right]f_{w^{\prime }}\left(z\right)+\left[z^{n}\right]f_{ww^{\prime }}\left(z\right)\right), \end{array}$$
where $f_{w,w^{\prime }}\left(z\right)=F_{w,w^{\prime }}\left(z,0,0\right)$ and $\left[z^{n}\right]F_{w,w^{\prime }}\left(z,0,0\right)=\left[z^{n}x_{1}^{0}x_{2}^{0}\right]F_{w,w^{\prime }}\left(z,x_{1},x_{2}\right)$. By (Equation 11) we have
(21)$$\begin{array}{cc} f_{w,w^{\prime }}\left(z\right) & =\frac{S_{w}\left(z\right)S_{w^{\prime }}\left(z\right)-S_{w,w^{\prime }}\left(z\right)S_{w^{\prime },w}\left(z\right)}{D_{w,w^{\prime }}\left(z\right)} \end{array}$$Having the above expression, we finally obtain
(22)$$\begin{array}{cc} G_{k}\left(z\right) & =\sum_{n\geq 0}E\left[{\left(X_{n,k}\right)}_{2}\right]z^{n}\\ \; & =\sum_{\begin{array}{c} w,w^{{}^{\prime }}\in A^{k}\\ w\neq w^{\prime } \end{array}}\sum_{n\geq 0}\left(1-\left[z^{n}\right]f_{w}\left(z\right)-\left[z^{n}\right]f_{w^{{}^{\prime }}}\left(z\right)+\left[z^{n}\right]f_{w,w^{{}^{\prime }}}\left(z\right)\right)z^{n}\\ \; & =\sum_{\begin{array}{c} w,w^{{}^{\prime }}\in A^{k}\\ w\neq w^{\prime } \end{array}}\left(\frac{1}{1-z}-f_{w}\left(z\right)-f_{w^{{}^{\prime }}}\left(z\right)+f_{w,w^{{}^{\prime }}}\left(z\right)\right)\\ \; & =\sum_{\begin{array}{c} w,w^{{}^{\prime }}\in A^{k}\\ w\neq w^{\prime } \end{array}}\left(\frac{1}{1-z}-\frac{S_{w}\left(z\right)}{D_{w}\left(z\right)}-\frac{S_{w^{{}^{\prime }}}\left(z\right)}{D_{w^{\prime }}\left(z\right)}+\frac{S_{w}\left(z\right)S_{w^{{}^{\prime }}}\left(z\right)-S_{w,w^{\prime }}\left(z\right)S_{w^{{}^{\prime }},w}\left(z\right)}{D_{w,w^{\prime }}\left(z\right)}\right). \end{array}$$ □

In the following lemma, we present the generating functions for the first two factorial moments for the *k*th Prefix Complexity in the independent model.

**Lemma** **4.**
*For ${\hat{H}}_{k}\left(z\right)=\sum_{n\geq 0}E\left[{\hat{X}}_{n,k}\right]z^{n}$ and ${\hat{G}}_{k}\left(z\right)=\sum_{n\geq 0}E\left[{\left({\hat{X}}_{n,k}\right)}_{2}\right]z^{n}$, which are the generating functions for $E[{\hat{X}}_{n,k}]$ and $E[{\left({\hat{X}}_{n,k}\right)}_{2}]$ respectively, we have*
*i.* 
(23)$$\begin{array}{cc} {\hat{H}}_{k}\left(z\right) & =\sum_{w\in A^{k}}\left(\frac{1}{1-z}-\frac{1}{1-(1-P(w))z}\right). \end{array}$$
*ii.* 
(24)$$\begin{array}{cc} {\hat{G}}_{k}\left(z\right) & =\sum_{\begin{array}{c} w,w^{{}^{\prime }}\in A^{k}\\ w\neq w^{\prime } \end{array}}\left(\frac{1}{1-z}-\frac{1}{1-(1-P(w))z}-\frac{1}{1-(1-P\left(w^{\prime }\right))z}\right)\\ \; & \;+\sum_{\begin{array}{c} w,w^{{}^{\prime }}\in A^{k}\\ w\neq w^{\prime } \end{array}}\frac{1}{1-(1-P\left(w\right)-P\left(w^{\prime }\right))z}. \end{array}$$



**Proof.** i. We define the indicator variable ${\hat{X}}_{n,k}^{\left(w\right)}$ as follows.
$${\hat{X}}_{n,k}^{\left(w\right)}=\left\{\begin{array}{cc} 1 & \;\mathrm{if}\;w\;\mathrm{is}\;a\;\mathrm{prefix}\;\mathrm{of}\;\mathrm{at}\;\mathrm{least}\;\mathrm{one}\;\mathrm{string}\;\mathrm{in}\;P\\ 0 & \;\mathrm{otherwise}. \end{array}\right.$$
For each ${\hat{X}}_{n,k}^{\left(w\right)}$, we have
(25)$$\begin{array}{cc} \;E[{\hat{X}}_{n,k}^{\left(w\right)}] & =P({\hat{X}}_{n,k}^{\left(w\right)}=1)\\ \; & =1-P({\hat{X}}_{n,k}^{\left(w\right)}=0)\\ \; & =1-{\left(1-P(w)\right)}^{n}. \end{array}$$
Summing over all words *w* of length *k*, determines the generating function $\hat{H}\left(z\right)$:
(26)$$\begin{array}{cc} \hat{H}\left(z\right) & =\sum_{n\geq 0}E\left[{\hat{X}}_{n,k}\right]z^{n}\\ \; & =\sum_{w\in A^{k}}\left(\frac{1}{1-z}-\frac{1}{1-(1-P(w))z}\right). \end{array}$$ii. Similar to in (Equation 18) and (Equation 20), we obtain
(27)$$\begin{array}{cc} E[{\left({\hat{X}}_{n,k}\right)}_{2}] & =\sum_{\begin{array}{c} w,w^{\prime }\in A^{k}\\ w\neq w^{\prime } \end{array}}E\left[{\hat{X}}_{n,k}^{\left(w\right)}{\hat{X}}_{n,k}^{\left(w^{\prime }\right)}\right]\\ \; & =\sum_{\begin{array}{c} w,w^{{}^{\prime }}\in A^{k}\\ w\neq w^{\prime } \end{array}}\left(1-{\left(1-P\left(w\right)\right)}^{n}-{\left(1-P\left(w^{\prime }\right)\right)}^{n}+{\left(1-P\left(w\right)-P\left(w^{\prime }\right)\right)}^{n}\right). \end{array}$$
Subsequently, we obtain the generating function below.
(28)$$\begin{array}{cc} \hat{G}\left(z\right) & =\sum_{n\geq 0}E\left[{\left({\hat{X}}_{n,k}\right)}_{2}\right]z^{n}\\ \; & =\sum_{\begin{array}{c} w,w^{{}^{\prime }}\in A^{k}\\ w\neq w^{\prime } \end{array}}\sum_{n\geq 0}\left(1-{\left(1-P\left(w\right)\right)}^{n}-{\left(1-P\left(w^{\prime }\right)\right)}^{n}+{\left(1-P\left(w\right)-P\left(w^{\prime }\right)\right)}^{n}\right)z^{n}\\ \; & =\sum_{\begin{array}{c} w,w^{{}^{\prime }}\in A^{k}\\ w\neq w^{\prime } \end{array}}\left(\frac{1}{1-z}-\frac{1}{1-(1-P(w))z}-\frac{1}{1-(1-P\left(w^{\prime }\right))z}\right)\\ \; & \;+\sum_{\begin{array}{c} w,w^{{}^{\prime }}\in A^{k}\\ w\neq w^{\prime } \end{array}}\frac{1}{1-(1-P\left(w\right)-P\left(w^{\prime }\right))z}. \end{array}$$ □

Our first goal is to compare the coefficients of the generating functions in the two models. The coefficients are expected to be asymptotically equivalent in the desired range for *k*. To compare the coefficients, we need more information on the analytic properties of these generating functions. This will be discussed in Section 3.3.

### 3.3. Analytic Properties of the Generating Functions

Here, we turn our attention to the smallest singularities of the two generating functions given in Lemma 3. It has been shown by Jacquet and Szpankowski [21] that $D_{w}\left(z\right)$ has exactly one root in the disk |z|≤ρ. Following the notations in [21], we denote the root within the disk |z|≤ρ of $D_{w}\left(z\right)$ by $A_{w}$, and by bootstrapping we obtain
(29)$$\begin{array}{cc} \; & A_{w}=1+\frac{1}{S_{w}\left(1\right)}P\left(w\right)+O\left(P{\left(w\right)}^{2}\right). \end{array}$$

We also denote the derivative of $D_{w}\left(z\right)$ at the root $A_{w}$, by $B_{w}$, and we obtain
(30)$$\begin{array}{c} B_{w}=-S_{w}\left(1\right)+\left(k-\frac{2S_{w}^{\prime }\left(1\right)}{S_{w}\left(1\right)}P\left(w\right)\right)+O\left(P{\left(w\right)}^{2}\right). \end{array}$$

In this paper, we will prove a similar result for the polynomial $D_{w,w^{\prime }}\left(z\right)$ through the following work.

**Lemma** **5.**
*If w and $w^{\prime }$ are two distinct binary words of length k and $\delta =\sqrt{p}$, there exists ρ>1, such that ρδ<1 and*
(31)$$\begin{array}{c} \sum_{w\in A^{k}}\left[\;[\right|S_{w,w^{\prime }}{\left(\rho \right)|\leq \left(\rho \delta \right)}^{k}\theta ]\;]P\left(w\right)\geq 1-\theta \delta^{k}. \end{array}$$


**Proof.** If the minimal degree of $S_{w,w^{\prime }}\left(z\right)$ is greater than >⌊k/2⌋, then
(32)$$\begin{array}{c} |S_{w,w^{\prime }}{\left(\rho \right)|\leq \left(\rho \delta \right)}^{k}\theta . \end{array}$$
for $\theta ={\left(1-p\right)}^{-1}$. For a fixed $w^{\prime }$, we have
(33)$$\begin{array}{cc} \; & \sum_{w\in A^{k}}\left[\;\left[S_{w,w^{\prime }}\left(z\right)\;\mathrm{has}\;\mathrm{minimal}\;\mathrm{degree}\leq \left\lfloor k/2\right\rfloor \;\right]\;\right]\;P\left(w\right)\\ \; & \;\;=\sum_{i=1}^{\left\lfloor k/2\right\rfloor }\sum_{w\in A^{k}}\left[\;\left[S_{w,w^{\prime }}\left(z\right)\;\mathrm{has}\;\mathrm{minimal}\;\mathrm{degree}\;=i\right]\;\right]\;P\left(w\right)\\ \; & \;\;=\sum_{i=1}^{\left\lfloor k/2\right\rfloor }\sum_{w_{1}...w_{i}\in A^{i}}P\left(w_{1}...w_{i}\right)\\ \; & \;\;\;\sum_{w_{i+1}...w_{k}\in A^{k-i}}\left[\;\left[S_{w,w^{\prime }}\left(z\right)\;\mathrm{has}\;\mathrm{minimal}\;\mathrm{degree}\;=i\right]\;\right]P\left(w_{i+1}...w_{k}\right)\\ \; & \;\;\leq \sum_{i=1}^{\left\lfloor k/2\right\rfloor }\sum_{w_{1}..w_{i}\in A^{i}}P\left(w_{i+1}...w_{k}\right)p^{k-i}\\ \; & \;\;=\sum_{i=1}^{\left\lfloor k/2\right\rfloor }p^{k-i}\sum_{w_{1}..w_{i}\in A^{i}}P\left(w_{1}...w_{i}\right)\\ \; & \;\;=\sum_{i=1}^{\left\lfloor k/2\right\rfloor }p^{k-i}\leq \frac{p^{k-\lfloor k/2\rfloor }}{1-p}. \end{array}$$This leads to the following
(34)$$\begin{array}{cc} \; & \sum_{w\in A^{k}}\left[\;\left[\;\mathrm{every}\;\mathrm{term}\;\mathrm{of}\;S_{w,w^{\prime }}\left(z\right)\;\mathrm{is}\;\mathrm{of}\;\mathrm{degree}>\left\lfloor k/2\right\rfloor \right]\;\right]P\left(w\right)\\ \; & \;\;=1-\sum_{w\in A^{k}}\left[\;\left[S_{w,w^{\prime }}\left(z\right)\;\mathrm{has}\;a\;\mathrm{term}\;\mathrm{of}\;\mathrm{degree}\leq \left\lfloor k/2\right\rfloor \right]\;\right]P\left(w\right)\\ \; & \;\;\geq 1-\frac{p^{\left\lceil k/2\right\rceil }}{1-p}\geq 1-\theta \delta^{k}. \end{array}$$ □

**Lemma** **6.**
*There exist $K^{\prime }>0$, and ρ>1 such that pρ<1, and such that, for every pair of distinct words w, and $w^{\prime }$ of length $k\geq K^{\prime }$, and for |z|≤ρ, we have*
(35)$$\begin{array}{c} |S_{w}\left(z\right)S_{w^{\prime }}\left(z\right)-S_{w,w^{\prime }}\left(z\right)S_{w^{\prime },w}\left(z\right)|>0. \end{array}$$

*In other words, $S_{w}\left(z\right)S_{w^{\prime }}\left(z\right)-S_{w,w^{\prime }}\left(z\right)S_{w^{\prime },w}\left(z\right)$ does not have any roots in |z|≤ρ.*


**Proof.** There are three cases to consider:Case i. When either $S_{w}\left(z\right)=1$ or $S_{w^{\prime }}\left(z\right)=1$, then every term of $S_{w,w^{\prime }}\left(z\right)S_{w^{\prime },w}\left(z\right)$ has degree *k* or larger, and therefore
(36)$$\begin{array}{c} |S_{w,w^{\prime }}\left(z\right)S_{w^{\prime },w}\left(z\right)|\leq k\frac{{\left(p\rho \right)}^{k}}{1-p\rho }. \end{array}$$
There exists $K_{1}>0$, such that for $k>K_{1}$, we have $\lim_{k\to \infty }k\frac{{\left(p\rho \right)}^{k}}{1-p\rho }=0$. This yields
(37)$$\begin{array}{cc} |S_{w}\left(z\right)S_{w^{\prime }}\left(z\right)-S_{w,w^{\prime }}\left(z\right)S_{w^{\prime },w}\left(z\right)| & \geq |S_{w}\left(z\right)S_{w^{\prime }}\left(z\right)|-|S_{w,w^{\prime }}\left(z\right)S_{w^{\prime },w}\left(z\right)|\\ \; & \geq 1-k\frac{{\left(p\rho \right)}^{k}}{1-p\rho }>0. \end{array}$$Case ii. If the minimal degree for $S_{w}\left(z\right)-1$ or $S_{w^{\prime }}\left(z\right)-1$ is greater than ⌊k/2⌋, then every term of $S_{w,w^{\prime }}\left(z\right)S_{w^{\prime },w}\left(z\right)$ has degree at least k/2. We also note that, by Lemma 9, $|S_{w}\left(z\right)S_{w^{\prime }}\left(z\right)|>0$. Therefore, there exists $K_{2}>0$, such that
(38)$$\begin{array}{cc} |S_{w}\left(z\right)S_{w^{\prime }}\left(z\right)-S_{w^{\prime },w}\left(z\right)S_{w,w^{\prime }}\left(z\right)| & \geq |S_{w}\left(z\right)S_{w^{\prime }}\left(z\right)|-|S_{w^{\prime },w}\left(z\right)S_{w,w^{\prime }}\left(z\right)|\\ \; & >0\;\mathrm{for}\;k>K_{2}. \end{array}$$Case iii. The only remaining case is where the minimal degree for $S_{w}\left(z\right)-1$ and $S_{w^{\prime }}\left(z\right)-1$ are both less than or equal to ⌊k/2⌋. If $w=w_{1}...w_{k}$, then $w^{\prime }=uw_{1}...w_{k-m}$, where *u* is a word of length m≥1. Then we have
(39)$$\begin{array}{c} S_{w^{\prime },w}\left(z\right)=P\left(w_{k-m+1}...w_{k}\right)z^{m}\left(S_{w}\left(z\right)-O\left({\left(pz\right)}^{k-m}\right)\right). \end{array}$$
There exists $K_{3}>0$, such that
(40)$$\begin{array}{cc} |S_{w^{\prime },w}\left(z\right)| & \leq {\left(p\rho \right)}^{m}\left(|S_{w}(z\right)|+O\left({\left(p\rho \right)}^{k-m}\right)\\ \; & ={\left(p\rho \right)}^{m}\left|S_{w}\left(z\right)\right|+O\left({\left(p\rho \right)}^{k}\right)\\ \; & <|S_{w}\left(z\right)|\;\mathrm{for}\;k>K_{3}\;. \end{array}$$Similarly, we can show that there exists $K_{3}^{\prime }$, such that $|S_{w,w^{\prime }}\left(z\right)|<|S_{w^{\prime }}\left(z\right)|$. Therefore, for $k>K_{3}^{\prime }$ we have
(41)$$\begin{array}{cc} |S_{w}\left(z\right)S_{w^{\prime }}\left(z\right)-S_{w,w^{\prime }}\left(z\right)S_{w^{\prime },w}\left(z\right)| & \geq |S_{w}\left(z\right)||S_{w^{\prime }}\left(z\right)|-|S_{w,w^{\prime }}\left(z\right)||S_{w^{\prime },w}\left(z\right)|\\ \; & >|S_{w}\left(z\right)||S_{w^{\prime }}\left(z\right)|-|S_{w}\left(z\right)||S_{w^{\prime }}\left(z\right)|=0. \end{array}$$We complete the proof by setting $K^{\prime }=\max\left\{K_{1},K_{2},K_{3},K_{3}^{\prime }\right\}$. □

**Lemma** **7.**
*There exist $K_{w,w^{\prime }}>0$ and ρ>1 such that pρ<1, and for every word w and $w^{\prime }$ of length $k\geq K_{w,w^{\prime }}$, the polynomial*
(42)$$\begin{array}{c} D_{w,w^{\prime }}\left(z\right)=\left(1-z\right)\left(S_{w}\left(z\right)S_{w^{\prime }}\left(z\right)-S_{w,w^{\prime }}\left(z\right)S_{w^{\prime },w}\left(z\right)\right)\\ +z^{k}\left(P\left(w\right)\left(S_{w^{\prime }}\left(z\right)-S_{w,w^{\prime }}\left(z\right)\right)+P\left(w^{\prime }\right)\left(S_{w}\left(z\right)-S_{w^{\prime },w}\left(z\right)\right)\right), \end{array}$$

*has exactly one root in the disk |z|≤ρ.*


**Proof.** First note that
(43)$$\begin{array}{cc} |S_{w}\left(z\right)-S_{w^{\prime },w}\left(z\right)| & \leq |S_{w}\left(z\right)|+|S_{w^{\prime },w}\left(z\right)|\\ \; & \leq \frac{1}{1-p\rho }+\frac{p\rho }{1-p\rho }=\frac{1+p\rho }{1-p\rho }. \end{array}$$This yields
(44)$$\begin{array}{cc} \; & \left|z^{k}\left(P\left(w\right)\left(S_{w^{\prime }}\left(z\right)-S_{w,w^{\prime }}\left(z\right)\right)+P\left(w^{\prime }\right)\left(S_{w}\left(z\right)-S_{w^{\prime },w}\left(z\right)\right)\right)\right|\\ \; & \;\;\leq {\left(p\rho \right)}^{k}\left(|S_{w}\left(z\right)-S_{w^{\prime },w}\left(z\right)|+|S_{w^{\prime }}\left(z\right)-S_{w,w^{\prime }}\left(z\right)|\right)\\ \; & \;\;\leq {\left(p\rho \right)}^{k}\left(\frac{2(1+p\rho )}{1-p\rho }\right). \end{array}$$There exist $K^{\prime }$, $K^{\prime \prime }$ large enough, such that, for $k>K^{\prime }$, we have
$$|(S_{w}\left(z\right)S_{w^{\prime }}\left(z\right)-S_{w,w^{\prime }}\left(z\right)S_{w^{\prime },w}\left(z\right))|\geq \beta >0,$$
and for $k>K^{\prime \prime }$,
$${\left(p\rho \right)}^{k}\left(\frac{2(1+p\rho )}{1-p\rho }\right)<\left(\rho -1\right)\beta .$$If we define $K_{w,w^{\prime }}=\max\left\{K^{\prime },K^{\prime \prime }\right\}$, then we have, for $k\geq K_{w,w^{\prime }}$,
(45)$$\begin{array}{cc} {\left(p\rho \right)}^{k}\left(\frac{2(1+p\rho )}{1-p\rho }\right) & <(\rho -1)\beta \\ \; & <|\left(1-z\right)(S_{w}\left(z\right)S_{w^{\prime }}\left(z\right)-S_{w,w^{\prime }}\left(z\right)S_{w^{\prime },w}\left(z\right))|. \end{array}$$
by Rouché’s theorem, as $\left(1-z\right)(S_{w}\left(z\right)S_{w^{\prime }}\left(z\right)-S_{w,w^{\prime }}\left(z\right)S_{w^{\prime },w}\left(z\right))$ has only one root in |z|≤ρ, then also $D_{w,w^{\prime }}\left(z\right)$ has exactly one root in |z|≤ρ. □

We denote the root within the disk |z|≤ρ of $D_{w,w^{\prime }}\left(z\right)$ by $\alpha_{w,w^{\prime }}$, and by bootstrapping we obtain
(46)$$\begin{array}{cc} \; & \alpha_{w,w^{\prime }}=1+\frac{S_{w^{\prime }}\left(1\right)-S_{w,w^{\prime }}\left(1\right)}{S_{w}\left(1\right)S_{w^{\prime }}\left(1\right)-S_{w,w^{\prime }}\left(1\right)S_{w^{\prime },w}\left(1\right)}P\left(w\right)\\ \; & \;\;+\frac{S_{w}\left(1\right)-S_{w^{\prime },w}\left(1\right)}{S_{w}\left(1\right)S_{w^{\prime }}\left(1\right)-S_{w,w^{\prime }}\left(1\right)S_{w^{\prime },w}\left(1\right)}P\left(w^{\prime }\right)+O\left(p^{2k}\right). \end{array}$$

We also denote the derivative of $D_{w,w^{\prime }}\left(z\right)$ at the root $\alpha_{w,w^{\prime }}$, by $\beta_{w,w^{\prime }}$, and we obtain
(47)$$\begin{array}{cc} \; & \beta_{w,w^{\prime }}=S_{w,w^{\prime }}\left(1\right)S_{w^{\prime },w}\left(1\right)-S_{w}\left(1\right)S_{w^{\prime }}\left(1\right)+O\left(kp^{k}\right). \end{array}$$

We will refer to these expressions in the residue analysis that we present in the next section.

### 3.4. Asymptotic Difference

We begin this section by the following lemmas on the autocorrelation polynomials.

**Lemma** **8**(Jacquet and Szpankowski, 1994). *For most words w, the autocorrelation polynomial $S_{w}\left(z\right)$ is very close to 1, with high probably. More precisely, if w is a binary word of length k and $\delta =\sqrt{p}$, there exists ρ>1, such that ρδ<1 and*
(48)$$\begin{array}{c} \sum_{w\in A^{k}}[\;[\left|S_{w}\left(\rho \right)-1\right|\leq {\left(\rho \delta \right)}^{k}\theta ]\;]P\left(w\right)\geq 1-\theta \delta^{k}, \end{array}$$
*where $\theta ={\left(1-p\right)}^{-1}$. We use Iverson notation*
$$\left[\;\left[A\right]\;\right]=\left\{\begin{array}{cc} 1 & if\;A\;holds\\ 0 & otherwise \end{array}\right.$$


**Lemma** **9**(Jacquet and Szpankowski, 1994). *There exist K>0 and ρ>1, such that pρ<1, and for every binary word w with length k≥K and |z|≤ρ, we have*
(49)$$\begin{array}{c} |S_{w}\left(z\right)|>0. \end{array}$$
*In other words, $S_{w}\left(z\right)$ does not have any roots in |z|≤ρ.*


**Lemma** **10.**
*With high probability, for most distinct pairs $\left\{w,w^{\prime }\right\}$, the correlation polynomial $S_{w,w^{\prime }}\left(z\right)$ is very close to 0. More precisely, if w and $w^{\prime }$ are two distinct binary words of length k and $\delta =\sqrt{p}$, there exists ρ>1, such that ρδ<1 and*
(50)$$\begin{array}{c} \sum_{w\in A^{k}}\left[\;[\right|S_{w,w^{\prime }}{\left(\rho \right)|\leq \left(\rho \delta \right)}^{k}\theta ]\;]P\left(w\right)\geq 1-\theta \delta^{k} \end{array}$$


We will use the above results to prove that the expected values in the Bernoulli model and the model built over a trie are asymptotically equivalent. We now prove Theorem 1 below.

**Proof** **of Theorem 1.**From Lemmas 3 and 4, we have
$$\begin{array}{c} H\left(z\right)=\sum_{w\in A^{k}}\left(\frac{1}{1-z}-\frac{S_{w}\left(z\right)}{D_{w}\left(z\right)}\right), \end{array}$$
and
$$\begin{array}{c} \hat{H}\left(z\right)=\sum_{w\in A^{k}}\left(\frac{1}{1-z}-\frac{1}{1-(1-P(w))z}\right). \end{array}$$
subtracting the two generating functions, we obtain
(51)$$\begin{array}{c} H\left(z\right)-\hat{H}\left(z\right)=\sum_{w\in A^{k}}\left(\frac{1}{1-(1-P(w))z}-\frac{S_{w}\left(z\right)}{D_{w}\left(z\right)}\right). \end{array}$$We define
(52)$$\begin{array}{c} \Delta_{w}\left(z\right)=\frac{1}{1-(1-P(w))z}-\frac{S_{w}\left(z\right)}{D_{w}\left(z\right)}. \end{array}$$Therefore, by Cauchy integral formula (see [20]), we have
(53)$$\begin{array}{cc} \left[z^{n}\right]\Delta_{w}\left(z\right) & =\frac{1}{2\pi i}\oint \Delta_{w}\left(z\right)\frac{dz}{z^{n+1}}={\mathrm{Res}}_{z=0}\;\Delta_{w}\left(z\right)\frac{dz}{z^{n+1}}, \end{array}$$
where the path of integration is a circle about zero with counterclockwise orientation. We note that the above integrand has poles at z=0, $z=\frac{1}{1-P(w)}$, and $z=A_{w}$ (refer to expression (Equation 29)). Therefore, we define
(54)$$\begin{array}{cc} I^{w}\left(\rho \right) & :=\frac{1}{2\pi i}\int_{|z|=\rho }\Delta_{w}\left(z\right)\frac{dz}{z^{n+1}}, \end{array}$$
where the circle of radius ρ contains all of the above poles. By the residue theorem, we have
(55)$$\begin{array}{cc} I^{w}\left(\rho \right) & ={\mathrm{Res}}_{z=0}\frac{\Delta_{w}\left(z\right)}{z^{n+1}}+{\mathrm{Res}}_{z=A_{w}}\frac{\Delta_{w}\left(z\right)}{z^{n+1}}+{\mathrm{Res}}_{z=1/1-P(w)}\frac{\Delta_{w}\left(z\right)}{z^{n+1}}\\ \; & =\left[z^{n}\right]\Delta_{w}\left(z\right)-{\mathrm{Res}}_{z=A_{w}}\frac{H_{w}\left(z\right)}{z^{n+1}}+{\mathrm{Res}}_{z=1/1-P(w)}\frac{{\hat{H}}_{w}\left(z\right)}{z^{n+1}} \end{array}$$We observe that
$$\begin{array}{cc} \; & {\mathrm{Res}}_{z=A_{w}}\frac{\Delta_{w}\left(z\right)}{z^{n+1}}=\frac{S_{w}\left(A_{w}\right)}{B_{w}\;A_{w}^{n+1}},\;\mathrm{where}\;B_{w}\;\mathrm{is}\;\mathrm{as}\;\mathrm{in}\;\;\left(30\right)\\ \; & {\mathrm{Res}}_{z=1/1-P(w)}\frac{{\hat{H}}_{w}\left(z\right)}{z^{n+1}}=-{\left(1-P\left(w\right)\right)}^{n+1}. \end{array}$$Then we obtain
(56)$$\begin{array}{cc} \left[z^{n}\right]\Delta_{w} & =I^{w}\left(\rho \right)-\frac{S_{w}\left(A_{w}\right)}{B_{w}\;A_{w}^{n+1}}-{\left(1-P\left(w\right)\right)}^{n+1}, \end{array}$$
and finally, we have
(57)$$\begin{array}{cc} \left[z^{n}\right]\left(H\left(z\right)-\hat{H}\left(z\right)\right) & =\sum_{w\in A^{k}}\left[z^{n}\right]\Delta_{w}\\ \; & =\sum_{w\in A^{k}}I_{n}^{w}\left(\rho \right)-\sum_{w\in A^{k}}\left(\frac{S_{w}\left(A_{w}\right)}{B_{w}\;A_{w}^{n+1}}+{\left(1-P\left(w\right)\right)}^{n+1}\right). \end{array}$$First, we show that, for sufficiently large *n*, the sum $\sum_{w\in A^{k}}\left(\frac{S_{w}\left(A_{w}\right)}{B_{w}\;A_{w}^{n+1}}+{\left(1-P\left(w\right)\right)}^{n+1}\right)$ approaches zero. □

**Lemma** **11.**
*For large enough n, and for k=Θ(logn), there exists M>0 such that*
(58)$$\begin{array}{c} \sum_{w\in A^{k}}\left(\frac{S_{w}\left(A_{w}\right)}{B_{w}\;A_{w}^{n+1}}+{\left(1-P\left(w\right)\right)}^{n+1}\right)=O\left(n^{-M}\right). \end{array}$$


**Proof.** We let
(59)$$\begin{array}{cc} r_{w}\left(z\right) & ={\left(1-P\left(w\right)\right)}^{z}+\frac{S_{w}\left(A_{w}\right)}{B_{w}A_{w}^{z}}. \end{array}$$The Mellin transform of the above function is
(60)$$\begin{array}{c} r_{w}^{*}\left(s\right)=\Gamma \left(s\right)\log^{-s}\left(\frac{1}{1-P(w)}\right)-\frac{S_{w}\left(A_{w}\right)}{B_{w}}\Gamma \left(s\right)\log^{-s}\left(A_{w}\right). \end{array}$$We define
(61)$$\begin{array}{cc} C_{w}=\frac{S_{w}\left(A_{w}\right)}{B_{w}}\; & =\frac{S_{w}\left(A_{w}\right)}{-S_{w}\left(1\right)+O\left(kP\left(w\right)\right)}, \end{array}$$
which is negative and uniformly bounded for all *w*. Also, for a fixed *s*, we have
(62)$$\begin{array}{cc} \ln^{-s}\left(\frac{1}{1-P(w)}\right) & =\ln^{-s}\left(1+P\left(w\right)+O\left(P{\left(w\right)}^{2}\right)\right)\\ \; & ={\left(P\left(w\right)+O\left(P{\left(w\right)}^{2}\right)\right)}^{-s}\\ \; & =P{\left(w\right)}^{-s}{\left(1+O\left(P(w)\right)\right)}^{-s}\\ \; & =P{\left(w\right)}^{-s}\left(1+O\left(P(w)\right)\right), \end{array}$$
(63)$$\begin{array}{cc} \ln^{-s}\left(A_{w}\right) & =\ln^{-s}\left(1-\left(-\frac{P(w)}{S_{w}\left(1\right)}+O\left(P{\left(w\right)}^{2}\right)\right)\right)\\ \; & ={\left(\frac{P(w)}{S_{w}\left(1\right)}+O\left(P{\left(w\right)}^{2}\right)\right)}^{-s}\\ \; & ={\left(\frac{P(w)}{S_{w}\left(1\right)}\right)}^{-s}{\left(1+O\left(P(w)\right)\right)}^{-s}\\ \; & ={\left(\frac{P(w)}{S_{w}\left(1\right)}\right)}^{-s}\left(1+O\left(P(w)\right)\right), \end{array}$$
and therefore, we obtain
(64)$$\begin{array}{cc} r_{w}^{*}\left(s\right) & =\Gamma \left(s\right)P{\left(w\right)}^{-s}\left(1-\frac{1}{S_{w}{\left(1\right)}^{-s}}\right)O\left(1\right). \end{array}$$From this expression, and noticing that the function has a removable singularity at s=0, we can see that the Mellin transform $r_{w}^{*}\left(s\right)$ exists on the strip where ℜ(s)>−1. We still need to investigate the Mellin strip for the sum $\sum_{w\in A^{k}}r_{w}^{*}\left(s\right)$. In other words, we need to examine whether summing $r_{w}^{*}\left(s\right)$ over all words of length *k* (where *k* grows with *n*) has any effect on the analyticity of the function. We observe that
$$\begin{array}{cc} \sum_{w\in A^{k}}\left|r_{w}^{*}\left(s\right)\right| & =\sum_{w\in A^{k}}\left|\Gamma \left(s\right)P{\left(w\right)}^{-s}\left(1-\frac{1}{S_{w}{\left(1\right)}^{-s}}\right)O\left(1\right)\right|\\ \; & \leq |\Gamma \left(s\right)|\;\sum_{w\in A^{k}}P{\left(w\right)}^{-ℜ(s)}\left(1-\frac{1}{S_{w}{\left(1\right)}^{-ℜ(s)}}\right)O\left(1\right)\\ \; & ={\left(q^{k}\right)}^{-ℜ(s)-1}\left|\Gamma \left(s\right)\right|\sum_{w\in A^{k}}P\left(w\right)\left(1-S_{w}{\left(1\right)}^{ℜ(s)}\right)O\left(1\right). \end{array}$$Lemma 8 allows us to split the above sum between the words for which $S_{w}\left(1\right)\leq 1+O\left(\delta^{k}\right)$ and words that have $S_{w}\left(1\right)>1+O\left(\delta^{k}\right)$.Such a split yields the following
(65)$$\begin{array}{cc} \sum_{w\in A^{k}}\left|r_{w}^{*}\left(s\right)\right| & ={\left(q^{k}\right)}^{-ℜ(s)-1}\left|\Gamma \left(s\right)\right|O\left(\delta^{k}\right). \end{array}$$This shows that $\sum_{w\in A^{k}}r_{w}^{*}\left(s\right)$ is bounded above for ℜ(s)>−1 and, therefore, it is analytic. This argument holds for k=Θ(logn) as well, as ${\left(q^{k}\right)}^{-ℜ(s)-1}$ would still be bounded above by a constant $M_{s,k}$ that depends on *s* and *k*.We would like to approximate $\sum_{w\in A^{k}}r_{w}^{*}\left(s\right)$ when z→∞. By the inverse Mellin transform, we have
(66)$$\begin{array}{cc} \sum_{w\in A^{k}}r_{w}\left(z\right) & =\frac{1}{2\pi i}\int_{c-i\infty }^{c+i\infty }\left(\sum_{w\in A^{k}}r_{w}^{*}\left(s\right)\right)z^{-s}ds. \end{array}$$We choose c∈(−1,M) for a fixed M>0. Then by the direct mapping theorem [22], we obtain
(67)$$\begin{array}{c} \sum_{w\in A^{k}}r_{w}\left(z\right)=O\left(z^{-M}\right). \end{array}$$
and subsequently, we get
(68)$$\begin{array}{c} \sum_{w\in A^{k}}\left(\frac{S_{w}\left(A_{w}\right)}{B_{w}\;A_{w}^{n+1}}+{\left(1-P\left(w\right)\right)}^{n+1}\right)=O\left(n^{-M}\right). \end{array}$$ □

We next prove the asymptotic smallness of $I_{n}^{w}\left(\rho \right)$ in (Equation 54).

**Lemma** **12.**
*Let*
(69)$$\begin{array}{cc} I_{n}^{w}\left(\rho \right)\; & =\frac{1}{2\pi i}\int_{|z|=\rho }\left(\frac{1}{1-(1-P(w))z}-\frac{S_{w}\left(z\right)}{D_{w}\left(z\right)}\right)\frac{dz}{z^{n+1}}. \end{array}$$

*For large n and k=Θ(logn), we have*
(70)$$\begin{array}{c} \sum_{w\in A^{k}}I_{n}^{w}\left(\rho \right)=O\left(\rho^{-n}{\left(\rho \delta \right)}^{k}\right). \end{array}$$


**Proof.** We observe that
(71)$$\begin{array}{c} |I_{n}^{w}\left(\rho \right)|\leq \frac{1}{2\pi }\int_{|z|=\rho }\left|\frac{P\left(w\right)z\left(z^{k-1}-S_{w}\left(z\right)\right)}{D_{w}\left(z\right)\left(1-\left(1-P\left(w\right)\right)z\right)}\frac{1}{z^{n+1}}\right|dz. \end{array}$$For |z|=ρ, we show that the denominator in (Equation 71) is bounded away from zero.
(72)$$\begin{array}{cc} |D_{w}\left(z\right)| & =|\left(1-z\right)S_{w}\left(z\right)+P\left(w\right)z^{k}|\\ \; & \geq \left|1-z|\right|S_{w}\left(z\right)|-P\left(w\right)|z^{k}|\\ \; & \geq \left(\rho -1\right)\alpha -{\left(p\rho \right)}^{k},\;\mathrm{where}\;\alpha >0\;\mathrm{by}\;\mathrm{Lemma}\;9\;.\\ \; & >0,\;\mathrm{we}\;\mathrm{assume}\;k\;\mathrm{to}\;\mathrm{be}\;\mathrm{large}\;\mathrm{enough}\;\mathrm{such}\;\mathrm{that}\;{\left(p\rho \right)}^{k}<\alpha \left(\rho -1\right). \end{array}$$To find a lower bound for |1−(1−P(w))z|, we can choose $K_{w}$ large enough such that
(73)$$\begin{array}{cc} \left|1-(1-P(w))z\right| & \geq \left|1-(1-P(w))|z|\right|\\ \; & \geq |1-\rho (1-p^{K_{w}})|\\ \; & >0. \end{array}$$We now move on to finding an upper bound for the numerator in (Equation 71), for |z|=ρ.
(74)$$\begin{array}{cc} |z^{k-1}-S_{w}\left(z\right)| & \leq |S_{w}\left(z\right)-1|+|1-z^{k-1}|\\ \; & \leq \left(S_{w}\left(\rho \right)-1\right)+\left(1+\rho^{k-1}\right)\\ \; & =\left(S_{w}\left(\rho \right)-1\right)+O\left(\rho^{k}\right). \end{array}$$Therefore, there exists a constant μ>0 such that
(75)$$\begin{array}{cc} |I_{n}^{w}| & \leq \mu \rho P\left(w\right)\left(\left(S_{w}\left(\rho \right)-1\right)+O\left(\rho^{k}\right)\right)\frac{1}{\rho^{n+1}}\\ \; & =O\left(\rho^{-n}\right)\left(P\left(w\right)\left(S_{w}\left(\rho \right)-1\right)+P\left(w\right)O\left(\rho^{k}\right)\right). \end{array}$$Summing over all patterns *w*, and applying Lemma 8, we obtain
(76)$$\begin{array}{cc} \sum_{w\in A^{k}}\left|I_{n}^{w}\left(\rho \right)\right| & =O\left(\rho^{-n}\right)\sum_{w\in A^{k}}P\left(w\right)\left(S_{w}\left(\rho \right)-1\right)+O\left(\rho^{-n+k}\right)\sum_{w\in A^{k}}P\left(w\right)\\ \; & =O\left(\rho^{-n}\right)\left(\theta {\left(\rho \delta \right)}^{k}+\frac{p\rho }{1-p\rho }\theta \delta^{k}\right)+O\left(\rho^{-n+k}\right)\\ \; & =O(\rho^{-n}{\left(\rho \delta \right)}^{k}), \end{array}$$
which approaches zero as n→∞ and k=Θ(logn). This completes the proof of of Theorem 1. □

Similar to Theorem 1, we provide a proof to show that the second factorial moments of the *k*th Subword Complexity and the *k*th Prefix Complexity, have the same first order asymptotic behavior. We are now ready to state the proof of Theorem 2.

**Proof** **of Theorem 2.**As discussed in Lemmas 3 and 4, the generating functions representing $E[{\left(X_{n,k}\right)}_{2}]$ and $E[{\left({\hat{X}}_{n,k}\right)}_{2}]$ respectively, are
$$\begin{array}{c} G\left(z\right)=\sum_{\begin{array}{c} w,w^{\prime }\in A^{k}\\ w\neq w^{\prime } \end{array}}\left(\frac{1}{1-z}-\frac{S_{w}\left(z\right)}{D_{w}\left(z\right)}-\frac{S_{w^{\prime }}\left(z\right)}{D_{w^{\prime }}\left(z\right)}+\frac{S_{w}\left(z\right)S_{w^{\prime }}\left(z\right)-S_{w,w^{\prime }}\left(z\right)S_{w^{\prime },w}\left(z\right)}{D_{w,w^{\prime }}\left(z\right)}\right), \end{array}$$
and
$$\begin{array}{cc} \hat{G}\left(z\right) & =\sum_{\begin{array}{c} w,w^{\prime }\in A^{k}\\ w\neq w^{\prime } \end{array}}\left(\frac{1}{1-z}-\frac{1}{1-(1-P(w))z}-\frac{1}{1-(1-P\left(w^{\prime }\right))z}\right)\\ \; & \;+\sum_{\begin{array}{c} w,w^{\prime }\in A^{k}\\ w\neq w^{\prime } \end{array}}\frac{1}{1-(1-P\left(w\right)-P\left(w^{\prime }\right))z}. \end{array}$$Note that
(77)$$\begin{array}{cc} G\left(z\right)-\hat{G}\left(z\right) & =\sum_{\begin{array}{c} w^{\prime }\in A^{k}\\ w\neq w^{\prime } \end{array}}\sum_{w\in A^{k}}\left(\frac{1}{1-(1-P(w))z}-\frac{S_{w}\left(z\right)}{D_{w}\left(z\right)}\right) \end{array}$$
(78)$$\begin{array}{cc} \; & +\sum_{\begin{array}{c} w\in A^{k}\\ w\neq w^{\prime } \end{array}}\sum_{w^{\prime }\in A^{k}}\left(\frac{1}{1-(1-P\left(w^{\prime }\right))z}-\frac{S_{w^{\prime }}\left(z\right)}{D_{w^{\prime }}\left(z\right)}\right) \end{array}$$
(79)$$\begin{array}{cc} \; & +\sum_{\begin{array}{c} w,w^{\prime }\in A^{k}\\ w\neq w^{\prime } \end{array}}\left(\frac{1}{1-(1-P\left(w\right)-P\left(w^{\prime }\right))z}-\frac{S_{w}\left(z\right)S_{w^{\prime }}\left(z\right)-S_{w,w^{\prime }}\left(z\right)S_{w^{\prime },w}\left(z\right)}{D_{w,w^{\prime }}\left(z\right)}\right) \end{array}$$In Theorem 1, we proved that for every M>0 (which does not depend on *n* or *k*), we have
$$\begin{array}{cc} H\left(z\right)-\hat{H}\left(z\right) & =\sum_{w\in A^{k}}\left(\frac{1}{1-(1-P(w))z}-\frac{S_{w}\left(z\right)}{D_{w}\left(z\right)}\right)=O\left(n^{-M}\right). \end{array}$$Therefore, both (Equation 77) and (Equation 78) are of order $\left(2^{k}-1\right)O\left(n^{-M}\right)=O\left(n^{-M+a\log2}\right)$ for k=alogn. Thus, to show the asymptotic smallness, it is enough to choose M=alog2+ϵ, where ϵ is a small positive value. Now, it only remains to show (Equation 79) is asymptotically negligible as well. We define
(80)$$\begin{array}{c} \Delta_{w,w^{\prime }}\left(z\right)=\frac{1}{1-(1-P\left(w\right)-P\left(w^{\prime }\right))z}-\frac{S_{w}\left(z\right)S_{w^{\prime }}\left(z\right)-S_{w,w^{\prime }}\left(z\right)S_{w^{\prime },w}\left(z\right)}{D_{w,w^{\prime }}\left(z\right)}. \end{array}$$
Next, we extract the coefficient of $z^{n}$
(81)$$\begin{array}{cc} \left[z^{n}\right]\Delta_{w,w^{\prime }}\left(z\right) & =\frac{1}{2\pi i}\oint \Delta_{w,w^{\prime }}\left(z\right)\frac{dz}{z^{n+1}}, \end{array}$$
where the path of integration is a circle about the origin with counterclockwise orientation. We define
(82)$$\begin{array}{c} I_{n}^{w,w^{\prime }}\left(\rho \right)=\frac{1}{2\pi i}\int_{|z|=\rho }\Delta_{w,w^{\prime }}\left(z\right)\frac{dz}{z^{n+1}}, \end{array}$$
The above integrand has poles at z=0, $z=\alpha_{w,w^{\prime }}$ (as in (Equation 46)), and $z=\frac{1}{1-P\left(w\right)-P\left(w^{\prime }\right)}$. We have chosen ρ such that the poles are all inside the circle |z|=ρ. It follows that
(83)$$\begin{array}{cc} I_{n}^{w,w^{\prime }}\left(\rho \right) & ={\mathrm{Res}}_{z=0}\frac{\Delta_{w,w^{\prime }}\left(z\right)}{z^{n+1}}+{\mathrm{Res}}_{z=\alpha_{w,w^{\prime }}}\frac{\Delta_{w,w^{\prime }}\left(z\right)}{z^{n+1}}+{\mathrm{Res}}_{z=\frac{1}{1-P\left(w\right)-P\left(w^{\prime }\right)}}\frac{\Delta_{w}\left(z\right)}{z^{n+1}}, \end{array}$$
and the residues give us the following.
$$\begin{array}{c} {\mathrm{Res}}_{z=\frac{1}{1-P\left(w\right)-P\left(w^{\prime }\right)}}\frac{1}{1-\left(1-P\left(w\right)-P\left(w^{\prime }\right)\right)z)z^{n+1}}=-{\left(1-P\left(w\right)-P\left(w^{\prime }\right)\right)}^{n+1}, \end{array}$$
and
$$\begin{array}{cc} \; & {\mathrm{Res}}_{z=\alpha_{w,w^{\prime }}}\frac{S_{w}\left(z\right)S_{w^{\prime }}\left(z\right)-S_{w,w^{\prime }}\left(z\right)S_{w^{\prime },w}\left(z\right)}{D_{w,w^{\prime }}\left(z\right)}=\\ \; & \;\;\;\;\;\;\frac{S_{w}\left(\alpha_{w,w^{\prime }}\right)S_{w^{\prime }}\left(\alpha_{w,w^{\prime }}\right)-S_{w,w^{\prime }}\left(\alpha_{w,w^{\prime }}\right)S_{w^{\prime },w}\left(\alpha_{w,w^{\prime }}\right)}{\beta_{w,w^{\prime }}\alpha_{w,w^{\prime }}^{n+1}}, \end{array}$$
where $\beta_{w,w^{\prime }}$ is as in (Equation 47). Therefore, we get
(84)$$\begin{array}{cc} \sum_{\begin{array}{c} w,w^{\prime }\in A^{k}\\ w\neq w^{\prime } \end{array}}\left[z^{n}\right]\Delta_{w,w^{\prime }}\left(z\right) & =\sum_{\begin{array}{c} w,w^{\prime }\in A^{k}\\ w\neq w^{\prime } \end{array}}I_{n}^{w,w^{\prime }}\left(\rho \right)\\ \; & \;-\sum_{\begin{array}{c} w,w^{\prime }\in A^{k}\\ w\neq w^{\prime } \end{array}}(\frac{S_{w}\left(\alpha_{w,w^{\prime }}\right)S_{w^{\prime }}\left(\alpha_{w,w^{\prime }}\right)-S_{w,w^{\prime }}\left(\alpha_{w,w^{\prime }}\right)S_{w^{\prime },w}\left(\alpha_{w,w^{\prime }}\right)}{\beta_{w,w^{\prime }}\alpha_{w,w^{\prime }}^{n+1}}\\ \; & \;\;\;\;+{\left(1-P\left(w\right)-P\left(w^{\prime }\right)\right)}^{n+1}). \end{array}$$
We now show that the above two terms are asymptotically small. □

**Lemma** **13.**
*There exists ϵ>0 where the sum*
$$\begin{array}{c} \sum_{\begin{array}{c} w,w^{\prime }\in A^{k}\\ w\neq w^{\prime } \end{array}}\left(\frac{S_{w}\left(\alpha_{w,w^{\prime }}\right)S_{w^{\prime }}\left(\alpha_{w,w^{\prime }}\right)-S_{w,w^{\prime }}\left(\alpha_{w,w^{\prime }}\right)S_{w^{\prime },w}\left(\alpha_{w,w^{\prime }}\right)}{\beta_{w,w^{\prime }}\alpha_{w,w^{\prime }}^{n+1}}+{\left(1-P\left(w\right)-P\left(w^{\prime }\right)\right)}^{n+1}\right) \end{array}$$

*is of order O($n^{-\epsilon }$).*


**Proof.** We define
$$\begin{array}{cc} r_{w,w^{\prime }}\left(z\right) & =\frac{S_{w}\left(\alpha_{w,w^{\prime }}\right)S_{w^{\prime }}\left(\alpha_{w,w^{\prime }}\right)-S_{w,w^{\prime }}\left(\alpha_{w,w^{\prime }}\right)S_{w^{\prime },w}\left(\alpha_{w,w^{\prime }}\right)}{\beta_{w,w^{\prime }}\alpha_{w,w^{\prime }}^{z}}+{\left(1-P\left(w\right)-P\left(w^{\prime }\right)\right)}^{z}. \end{array}$$
The Mellin transform of the above function is
(85)$$\begin{array}{c} r_{w,w^{\prime }}^{*}\left(s\right)=\Gamma \left(s\right)\log^{-s}\left(\frac{1}{1-P\left(w\right)-p\left(w^{\prime }\right)}\right)+C_{w,w^{\prime }}\Gamma \left(s\right)\log^{-s}\left(\alpha_{w,w^{\prime }}\right), \end{array}$$
where $C_{w,w^{\prime }}=\frac{S_{w}\left(\alpha_{w,w^{\prime }}\right)S_{w^{\prime }}\left(\alpha_{w,w^{\prime }}\right)-S_{w,w^{\prime }}\left(\alpha_{w,w^{\prime }}\right)S_{w^{\prime },w}\left(\alpha_{w,w^{\prime }}\right)}{\beta_{w,w^{\prime }}}$. We note that $C_{w,w^{\prime }}$ is negative and uniformly bounded from above for all $w,w^{\prime }\in A^{k}$.For a fixes *s*, we also have,
(86)$$\begin{array}{cc} \ln^{-s}\left(\frac{1}{1-P\left(w\right)-P\left(w^{\prime }\right)}\right) & =\ln^{-s}\left(1+P\left(w\right)+P\left(w^{\prime }\right)+O\left(p^{2k}\right)\right)\\ \; & ={\left(P\left(w\right)+P\left(w^{\prime }\right)+O\left(p^{2k}\right)\right)}^{-s}\\ \; & ={\left(P\left(w\right)+P\left(w^{\prime }\right)\right)}^{-s}{\left(1+O\left(p^{k}\right)\right)}^{-s}\\ \; & ={\left(P\left(w\right)+P\left(w^{\prime }\right)\right)}^{-s}\left(1+O\left(p^{k}\right)\right), \end{array}$$
and
(87)$$\begin{array}{cc} \ln^{-s}\left(\alpha_{w,w^{\prime }}\right) & =(\frac{S_{w^{\prime }}\left(1\right)-S_{w,w^{\prime }}\left(1\right)}{S_{w}\left(1\right)S_{w^{\prime }}\left(1\right)-S_{w,w^{\prime }}\left(1\right)S_{w^{\prime },w}\left(1\right)}P\left(w\right)\\ \; & \;\;+\frac{S_{w}\left(1\right)-S_{w^{\prime },w}\left(1\right)}{S_{w}\left(1\right)S_{w^{\prime }}\left(1\right)-S_{w,w^{\prime }}\left(1\right)S_{w^{\prime },w}\left(1\right)}P\left(w^{\prime }\right)+O\left(p^{2k}\right){)}^{-s}\\ \; & =(\frac{S_{w^{\prime }}\left(1\right)-S_{w,w^{\prime }}\left(1\right)}{S_{w}\left(1\right)S_{w^{\prime }}\left(1\right)-S_{w,w^{\prime }}\left(1\right)S_{w^{\prime },w}\left(1\right)}P\left(w\right)\\ \; & \;\;+\frac{S_{w}\left(1\right)-S_{w^{\prime },w}\left(1\right)}{S_{w}\left(1\right)S_{w^{\prime }}\left(1\right)-S_{w,w^{\prime }}\left(1\right)S_{w^{\prime },w}\left(1\right)}P\left(w^{\prime }\right){)}^{-s}\left(1+O(p^{k})\right). \end{array}$$
Therefore, we have
(88)$$\begin{array}{cc} r_{w,w^{\prime }}^{*}\left(s\right) & =\Gamma \left(s\right){\left(P\left(w\right)+P\left(w^{\prime }\right)\right)}^{-s}\left(1+O\left(p^{k}\right)\right)\\ \; & \;-\Gamma \left(s\right)(\frac{S_{w^{\prime }}\left(1\right)-S_{w,w^{\prime }}\left(1\right)}{S_{w}\left(1\right)S_{w^{\prime }}\left(1\right)-S_{w,w^{\prime }}\left(1\right)S_{w^{\prime },w}\left(1\right)}P\left(w\right)\\ \; & \;\;\;+\frac{S_{w}\left(1\right)-S_{w^{\prime },w}\left(1\right)}{S_{w}\left(1\right)S_{w^{\prime }}\left(1\right)-S_{w,w^{\prime }}\left(1\right)S_{w^{\prime },w}\left(1\right)}P\left(w^{\prime }\right){)}^{-s}\left(1+O(p^{k})\right)O\left(1\right). \end{array}$$
To find the Mellin strip for the sum $\sum_{w\in A^{k}}r_{w,w^{\prime }}^{*}\left(s\right)$, we first note that
$$\begin{array}{cc} {\left(x+y\right)}^{a} & \leq x^{a}+y^{a},\;\mathrm{for}\;\mathrm{any}\;\mathrm{real}\;x,y>0\;\mathrm{and}\;a\leq 1. \end{array}$$
Since −ℜ(s)<1, we have
(89)$$\begin{array}{c} {\left(P\left(w\right)+P\left(w^{\prime }\right)\right)}^{-ℜ(s)}\leq P{\left(w\right)}^{-ℜ(s)}+P{\left(w^{\prime }\right)}^{-ℜ(s)}, \end{array}$$
and
(90)$$\begin{array}{cc} \; & {\left(\frac{S_{w^{\prime }}\left(1\right)-S_{w,w^{\prime }}\left(1\right)}{S_{w}\left(1\right)S_{w^{\prime }}\left(1\right)-S_{w,w^{\prime }}\left(1\right)S_{w^{\prime },w}\left(1\right)}P\left(w\right)ın+\frac{S_{w}\left(1\right)-S_{w^{\prime },w}\left(1\right)}{S_{w}\left(1\right)S_{w^{\prime }}\left(1\right)-S_{w,w^{\prime }}\left(1\right)S_{w^{\prime },w}\left(1\right)}P\left(w^{\prime }\right)\right)}^{-ℜ(s)}\\ \; & \;\;\;\;\leq {\left(\frac{S_{w^{\prime }}\left(1\right)-S_{w,w^{\prime }}\left(1\right)}{S_{w}\left(1\right)S_{w^{\prime }}\left(1\right)-S_{w,w^{\prime }}\left(1\right)S_{w^{\prime },w}\left(1\right)}P\left(w\right)\right)}^{-ℜ(s)}\\ \; & \;\;\;\;\;\;\;+{\left(\frac{S_{w}\left(1\right)-S_{w^{\prime },w}\left(1\right)}{S_{w}\left(1\right)S_{w^{\prime }}\left(1\right)-S_{w,w^{\prime }}\left(1\right)S_{w^{\prime },w}\left(1\right)}P\left(w^{\prime }\right)\right)}^{-ℜ(s)}. \end{array}$$
Therefore, we get
$$\begin{array}{cc} \sum_{\begin{array}{c} w,w^{\prime }\in A^{k}\\ w\neq w^{\prime } \end{array}}\left|r_{w,w^{\prime }}^{*}\left(s\right)\right| & \leq |\Gamma (s)|O(1)\\ \; & \;(\sum_{\begin{array}{c} w,w^{\prime }\in A^{k}\\ w\neq w^{\prime } \end{array}}P{\left(w\right)}^{-ℜ(s)}\left(1-{\left(\frac{S_{w}\left(1\right)S_{w^{\prime }}\left(1\right)-S_{w,w^{\prime }}\left(1\right)S_{w^{\prime },w}\left(1\right)}{S_{w^{\prime }}\left(1\right)-S_{w,w^{\prime }}\left(1\right)}\right)}^{ℜ(s)}\right)\\ \; & +\sum_{\begin{array}{c} w,w^{\prime }\in A^{k}\\ w\neq w^{\prime } \end{array}}P{\left(w^{\prime }\right)}^{-ℜ(s)}\left(1-{\left(\frac{S_{w}\left(1\right)S_{w^{\prime }}\left(1\right)-S_{w,w^{\prime }}\left(1\right)S_{w^{\prime },w}\left(1\right)}{S_{w}\left(1\right)-S_{w^{\prime },w}\left(1\right)}\right)}^{ℜ(s)}\right))\\ \; & \leq {\left(q^{k}\right)}^{-ℜ(s)-1}\left|\Gamma \left(s\right)\right|O\left(1\right) \end{array}$$
(91)$$\begin{array}{cc} \; & (\sum_{\begin{array}{c} w^{\prime }\in A^{k}\\ w\neq w^{\prime } \end{array}}\sum_{w\in A^{k}}P\left(w\right)\left(1-{\left(S_{w}\left(1\right)\right)}^{ℜ(s)}{\left(1-\frac{S_{w,w^{\prime }}\left(1\right)}{S_{w^{\prime }}\left(1\right)}\right)}^{-ℜ(s)}\right) \end{array}$$
(92)$$\begin{array}{cc} \; & +\sum_{\begin{array}{c} w^{\prime }\in A^{k}\\ w\neq w^{\prime } \end{array}}\sum_{w\in A^{k}}P\left(w\right)S_{w,w^{\prime }}{\left(1\right)}^{ℜ(s)}{\left(\frac{S_{w^{\prime }}\left(1\right)-S_{w,w^{\prime }}\left(1\right)}{S_{w^{\prime },w}\left(1\right)}\right)}^{-ℜ(s)} \end{array}$$
(93)$$\begin{array}{cc} \; & +\sum_{\begin{array}{c} w\in A^{k}\\ w\neq w^{\prime } \end{array}}\sum_{w^{\prime }\in A^{k}}P\left(w^{\prime }\right)\left(1-{\left(S_{w^{\prime }}\left(1\right)\right)}^{ℜ(s)}{\left(1-\frac{S_{w^{\prime },w}\left(1\right)}{S_{w}\left(1\right)}\right)}^{-ℜ(s)}\right) \end{array}$$
(94)$$\begin{array}{cc} \; & +\sum_{\begin{array}{c} w\in A^{k}\\ w\neq w^{\prime } \end{array}}\sum_{w^{\prime }\in A^{k}}P\left(w^{\prime }\right)S_{w^{\prime },w}{\left(1\right)}^{ℜ(s)}{\left(\frac{S_{w}\left(1\right)-S_{w^{\prime },w}\left(1\right)}{S_{w,w^{\prime }}\left(1\right)}\right)}^{-ℜ(s)}). \end{array}$$
By Lemma 10, with high probability, a randomly selected *w* has the property $S_{w,w^{\prime }}\left(1\right)=O\left(\delta^{k}\right)$, and thus
$${\left(1-\frac{S_{w,w^{\prime }}\left(1\right)}{S_{w^{\prime }}\left(1\right)}\right)}^{-ℜ(s)}=1+O\left(\delta^{k}\right).$$
With that and by Lemma 8, for most words *w*,
$$1-S_{w}{\left(1\right)}^{ℜ(s)}\left(1+O\left(\delta^{k}\right)\right)=O\left(\delta^{k}\right).$$
Therefore, both sums (Equation 91) and (Equation 93) are of the form $\left(2^{k}-1\right)O\left(\delta^{k}\right)$. The sums (Equation 92) and (Equation 94) are also of order $\left(2^{k}-1\right)O\left(\delta^{k}\right)$ by Lemma 10. Combining all these terms we will obtain
(95)$$\begin{array}{cc} \sum_{\begin{array}{c} w,w^{\prime }\in A^{k}\\ w\neq w^{\prime } \end{array}}\left|r_{w,w^{\prime }}^{*}\left(s\right)\right| & \leq \left(2^{k}-1\right){\left(q^{k}\right)}^{-ℜ(s)-1}\left|\Gamma \left(s\right)\right|O\left(\delta^{k}\right)O\left(1\right). \end{array}$$
By the inverse Mellin transform, for k=alogn, M=alog2+ϵ and c∈(−1,M), we have
(96)$$\begin{array}{cc} \sum_{\begin{array}{c} w,w^{\prime }\in A^{k}\\ w\neq w^{\prime } \end{array}}r_{w,w^{\prime }}\left(z\right) & =\frac{1}{2\pi i}\int_{c-i\infty }^{c+i\infty }\left(\sum_{\begin{array}{c} w,w^{\prime }\in A^{k}\\ w\neq w^{\prime } \end{array}}r_{w,w^{\prime }}^{*}\left(s\right)\right)z^{-s}ds=O\left(z^{-M}\right)O\left(2^{k}\right)\\ \; & =O(z^{-\epsilon }). \end{array}$$ □

In the following lemma we show that the first term in (Equation 85) is asymptotically small.

**Lemma** **14.**
*Recall that*
$$\begin{array}{cc} I_{n}^{w,w^{\prime }}\left(\rho \right)\; & =\frac{1}{2\pi i}\int_{|z|=\rho }\Delta_{w,w^{\prime }}\left(z\right)\frac{dz}{z^{n+1}}. \end{array}$$

*We have*
(97)$$\begin{array}{cc} \sum_{\begin{array}{c} w,w^{\prime }\in A^{k}\\ w\neq w^{\prime } \end{array}}I_{n}^{w,w^{\prime }}\left(\rho \right) & =O\left(\rho^{-n+2k}\delta^{k}\right). \end{array}$$


**Proof.** First note that
(98)$$\begin{array}{cc} \Delta_{w,w^{\prime }}\left(z\right) & =\frac{1}{1-(1-P\left(w\right)-P\left(w^{\prime }\right))z}-\frac{S_{w}\left(z\right)S_{w^{\prime }}\left(z\right)-S_{w,w^{\prime }}\left(z\right)S_{w^{\prime },w}\left(z\right)}{D_{w,w^{\prime }}\left(z\right)}\\ \; & =\frac{zP\left(w\right)\left(S_{w,w^{\prime }}\left(z\right)S_{w^{\prime },w}\left(z\right)-S_{w}\left(z\right)S_{w^{\prime }}\left(z\right)+z^{k-1}S_{w^{\prime }}\left(z\right)-z^{k-1}S_{w,w^{\prime }}\left(z\right)\right)}{\left(1-(1-P\left(w\right)-P\left(w^{\prime }\right))z\right)D_{w,w^{\prime }}\left(z\right)}\\ \; & \;\;+\frac{zP\left(w^{\prime }\right)\left(S_{w^{\prime },w}\left(z\right)S_{w,w^{\prime }}\left(z\right)-S_{w^{\prime }}\left(z\right)S_{w}\left(z\right)+z^{k-1}S_{w}\left(z\right)-z^{k-1}S_{w^{\prime },w}\left(z\right)\right)}{\left(1-(1-P\left(w\right)-P\left(w^{\prime }\right))z\right)D_{w,w^{\prime }}\left(z\right)}. \end{array}$$
We saw in (Equation 73) that $|1-\left(1-P\left(w^{\prime }\right)\right)z|\geq c_{2}$, and therefore, it follows that
(99)$$\begin{array}{c} |1-\left(1-P\left(w\right)-P\left(w^{\prime }\right)\right)z|\geq c_{1} \end{array}$$
For z=ρ, $|D_{w,w^{\prime }}\left(z\right)|$ is also bounded below as the following
(100)$$\begin{array}{cc} |D_{w,w^{\prime }}\left(z\right)| & =|\left(1-z\right)(S_{w}\left(z\right)S_{w^{\prime }}\left(z\right)-S_{w,w^{\prime }}\left(z\right)S_{w^{\prime },w}\left(z\right))\\ \; & \;\;+z^{k}\left(P\left(w\right)\left(S_{w^{\prime }}\left(z\right)-S_{w,w^{\prime }}\left(z\right)\right)+P\left(w^{\prime }\right)\left(S_{w}\left(z\right)-S_{w^{\prime },w}\left(z\right)\right)\right)|\\ \; & \geq |\left(1-z\right)(S_{w}\left(z\right)S_{w^{\prime }}\left(z\right)-S_{w,w^{\prime }}\left(z\right)S_{w^{\prime },w}\left(z\right))|\\ \; & \;\;-\left|z^{k}\right|\left|\left(P\left(w\right)\left(S_{w^{\prime }}\left(z\right)-S_{w,w^{\prime }}\left(z\right)\right)+P\left(w^{\prime }\right)\left(S_{w}\left(z\right)-S_{w^{\prime },w}\left(z\right)\right)\right)\right|\\ \; & \geq \left(\rho -1\right)\beta -{\left(p\rho \right)}^{k}\left(\frac{2(1+p\rho )}{1-p\rho }\right), \end{array}$$
which is bounded away from zero by the assumption of Lemma 7. Additionally, we show that the numerator in (Equation 98) is bounded above, as follows
(101)$$\begin{array}{cc} \; & |S_{w,w^{\prime }}\left(z\right)S_{w^{\prime },w}\left(z\right)-S_{w}\left(z\right)S_{w^{\prime }}\left(z\right)+z^{k-1}S_{w^{\prime }}\left(z\right)-z^{k-1}S_{w,w^{\prime }}\left(z\right)|\leq \\ \; & \;|S_{w^{\prime }}\left(z\right)\left(z^{k-1}-S_{w}\left(z\right)\right)\left|+\right|S_{w,w^{\prime }}\left(z\right)\left(S_{w^{\prime },w}\left(z\right)-z^{k-1}\right)|\\ \; & \;\leq S_{w^{\prime }}\left(\rho \right)\left(\left(S_{w}\left(\rho \right)-1\right)+O\left(\rho^{k}\right)\right)+S_{w,w^{\prime }}\left(\rho \right)\left(S_{w^{\prime },w}\left(\rho \right)+O\left(\rho^{k}\right)\right). \end{array}$$
This yields
(102)$$\begin{array}{cc} \sum_{\begin{array}{c} w,w^{\prime }\in A^{k}\\ w\neq w^{\prime } \end{array}}\left|I_{n}^{w,w^{\prime }}\right| & \leq O\left(\rho^{-n}\right)\sum_{\begin{array}{c} w^{\prime }\in A^{k}\\ w\neq w^{\prime } \end{array}}S_{w^{\prime }}\left(\rho \right)\sum_{w\in A^{k}}P\left(w\right)\left(\left(S_{w}\left(\rho \right)-1\right)+O\left(\rho^{k}\right)\right)\\ \; & \;+O\left(\rho^{-n}\right)\sum_{\begin{array}{c} w^{\prime }\in A^{k}\\ w\neq w^{\prime } \end{array}}\sum_{w\in A^{k}}P\left(w\right)S_{w,w^{\prime }}\left(\rho \right)\left(S_{w^{\prime },w}\left(\rho \right)+O\left(\rho^{k}\right)\right). \end{array}$$
By (Equation 75), the first term above is of order $\left(2^{k}-1\right)O\left(\rho^{-n+k}\right)$ and by Lemma 10 and an analysis similar to (Equation 75), the second term yields $\left(2^{k}-1\right)O\left(\rho^{-n+k}\right)$ as well. Finally, we have
$$\begin{array}{cc} \sum_{\begin{array}{c} w,w^{\prime }\in A^{k}\\ w\neq w^{\prime } \end{array}}\left|I_{n}^{w,w^{\prime }}\right| & \leq O(\rho^{-n+2k}\delta^{k}). \end{array}$$
Which goes to zero asymptotically, for k=Θ(logn). □

This lemma completes our proof of Theorem 2.

### 3.5. Asymptotic Analysis of the *k*th Prefix Complexity

We finally proceed to analyzing the asymptotic moments of the *k*th Prefix Complexity. The results obtained hold true for the moments of the *k*th Subword Complexity. Our methodology involves poissonization, saddle point analysis (the complex version of Laplace’s method [23]), and depoissonization.

**Lemma** **15**(Jacquet and Szpankowski, 1998). *Let $\tilde{G}\left(z\right)$ be the Poisson transform of a sequence $g_{n}$. If $\tilde{G}\left(z\right)$ is analytic in a linear cone $S_{\theta }$ with θ<π/2, and if the following two conditions hold:*
*(I) For $z\in S_{\theta }$ and real values B, r>0, ν*
(103)$$\begin{array}{c} \left|z|>r\to \right|\tilde{G}\left(z\right)|\leq B|z^{\nu }|\Psi (|z|), \end{array}$$

*where Ψ(x) is such that, for fixed t, $\lim_{x\to \infty }\frac{\Psi (tx)}{\Psi (x)}=1$;*

*(II) For $z\notin S_{\theta }$ and A,α<1*
(104)$$\begin{array}{c} \left|z|>r\to \right|\tilde{G}\left(z\right)e^{z}|\leq Ae^{\alpha |z|}. \end{array}$$
*Then, for every non-negative integer n, we have*
$$g_{n}=\tilde{G}\left(n\right)+O\left(n^{\nu -1}\Psi \left(n\right)\right).$$


**On the Expected Value:** To transform the sequence of interest, ${\left(E\left[{\hat{X}}_{n,k}\right]\right)}_{n\geq 0}$, into a Poisson model, we recall that in (Equation 25) we found
$$\begin{array}{cc} E[{\hat{X}}_{n,k}]\; & =\sum_{w\in A^{k}}\left(1-{\left(1-P(w)\right)}^{n}\right). \end{array}$$
Thus, the Poisson transform is
(105)$$\begin{array}{cc} {\tilde{E}}_{k}\left(z\right) & =\sum_{n=0}^{\infty }E\left[{\hat{X}}_{n,k}\right]\frac{z^{n}}{n!}e^{-z}\\ \; & =\sum_{n=0}^{\infty }\sum_{w\in A^{k}}\left(1-{\left(1-P\left(w\right)\right)}^{n}\right)\frac{z^{n}}{n!}e^{-z}\\ \; & =\sum_{w\in A^{k}}\left(1-e^{-zP(w)}\right). \end{array}$$
To asymptotically evaluate this harmonic sum, we turn our attention to the Mellin Transform once more. The Mellin transform of ${\tilde{E}}_{k}\left(z\right)$ is
(106)$$\begin{array}{cc} {\tilde{E}}_{k}^{*}\left(s\right) & =-\Gamma \left(s\right)\sum_{w\in A^{k}}P{\left(w\right)}^{-s}\\ \; & =-\Gamma \left(s\right){\left(p^{-s}+q^{-s}\right)}^{k}, \end{array}$$
which has the fundamental strip s∈〈−1,0〉. For c∈(−1,0), the inverse Mellin integral is the following
(107)$$\begin{array}{cc} \;{\tilde{E}}_{k}\left(z\right) & =\frac{1}{2\pi i}\int_{c-i\infty }^{c+i\infty }{\tilde{E}}_{k}^{*}\left(s\right)\cdot z^{-s}ds\\ \; & =\frac{-1}{2\pi i}\int_{c-i\infty }^{c+i\infty }z^{-s}\Gamma \left(s\right){\left(p^{-s}+q^{-s}\right)}^{k}ds\\ \; & =\frac{-1}{2\pi i}\int_{c-i\infty }^{c+i\infty }\Gamma \left(s\right)e^{-k(s\frac{\log z}{k}-\log\left(p^{-s}+q^{-s}\right))}ds\\ \; & =\frac{-1}{2\pi i}\int_{c-i\infty }^{c+i\infty }\Gamma \left(s\right)e^{-kh(s)}ds, \end{array}$$
where we define $h\left(s\right)=\frac{s}{a}-\log\left(p^{-s}+q^{-s}\right)$ for k=alogz. We emphasize that the above integral involves *k*, and *k* grows with *n*. We evaluate the integral through the saddle point analysis. Therefore, we choose the line of integration to cross the saddle point $r_{0}$. To find the saddle point $r_{0}$, we let $h^{\prime }\left(r_{0}\right)=0$, and we obtain
(108)$$\begin{array}{c} {\left(p/q\right)}^{-r_{0}}=\frac{a\log p^{-1}-1}{1-a\log q^{-1}}, \end{array}$$
and therefore,
(109)$$\begin{array}{cc} \; & r_{0}=\frac{-1}{\log p/q}\log\left(\frac{a\log q^{-1}-1}{1-a\log p^{-1}}\right), \end{array}$$
where $\frac{1}{\log q^{-1}}<a<\frac{1}{\log p^{-1}}$.

By (Equation 108) and the fact that ${\left(p/q\right)}^{it_{j}}=1$ for $t_{j}=\frac{2\pi j}{\log p/q}$ and j∈Z, we can see that there are actually infinitely many saddle points $z_{j}$ of the form $r_{0}+it_{j}$ on the line of integration.

We remark that the location of $r_{0}$ depends on the value of *a*. We have $r_{0}\to \infty$ as $a\to \frac{1}{\log q^{-1}}$, and $r_{0}\to -\infty$ as $a\to \frac{1}{\log p^{-1}}$. We divide the analysis into three parts, for the three ranges $r_{0}\in \left(0,\infty \right)$, $r_{0}\in \left(-1,0\right)$, and $r_{0}\in \left(-\infty ,-1\right)$.

In the first range, which corresponds to
(110)$$\begin{array}{c} \frac{1}{\log q^{-1}}<a<\frac{2}{\log q^{-1}+\log p^{-1}}, \end{array}$$
we perform a residue analysis, taking into account the dominant pole at s=−1. In the second range, we have
(111)$$\begin{array}{c} \frac{2}{\log q^{-1}+\log p^{-1}}<a<\frac{1}{q\log q^{-1}+p\log p^{-1}}, \end{array}$$
and we get the asymptotic result through the saddle point method. The last range corresponds to
(112)$$\begin{array}{c} \frac{1}{q\log q^{-1}+p\log p^{-1}}<a<\frac{1}{\log p^{-1}}, \end{array}$$
and we approach it with a combination of residue analysis at s=0, and the saddle point method. We now proceed by stating the proof of Theorem 3.

**Proof** **of Theorem 3.**We begin with proving part ii which requires a saddle point analysis. We rewrite the inverse Mellin transform with integration line at $ℜ\left(s\right)=r_{0}$ as
(113)$$\begin{array}{cc} {\tilde{E}}_{k}\left(z\right) & =\frac{-1}{2\pi }\int_{-\infty }^{\infty }z^{-(r_{0}+it)}\Gamma \left(r_{0}+it\right){\left(p^{-(r_{0}+it)}+q^{-(r_{0}+it)}\right)}^{k}dt\\ \; & =\frac{-1}{2\pi }\int_{-\infty }^{\infty }\Gamma \left(r_{0}+it\right)e^{-k(\left(r_{0}+it\right)\frac{\log z}{k}-\log\left(p^{-(r_{0}+it)}+q^{-(r_{0}+it)}\right))}dt. \end{array}$$
**Step one: Saddle points’ contribute to the integral estimation**
First, we are able to show those saddle points with $|t_{j}|>\sqrt{\log n}$ do not have a significant asymptotic contribution to the integral. To show this, we let
(114)$$\begin{array}{cc} T_{k}\left(z\right) & =\int_{|t|>\sqrt{\log n}}z^{-r_{0}-it}\Gamma \left(r_{0}+it\right){\left(p^{-r_{0}-it}+q^{-r_{0}-it}\right)}^{k}dt. \end{array}$$
Since $|\Gamma \left(r_{0}+it\right){\left|=O(|t\right|}^{r_{0}-\frac{1}{2}}e^{\frac{-\pi |t|}{2}})$ as |t|→±∞, we observe that
(115)$$\begin{array}{cc} T_{k}\left(z\right) & =O\left(z^{-r_{0}}{\left(p^{-r_{0}}+q^{-r_{0}}\right)}^{k}\int_{\sqrt{\log n}}^{\infty }t^{r_{0}/2-1/2}e^{-\pi t/2}dt\right)\\ \; & =O\left(z^{-r_{0}}{\left(p^{-r_{0}}+q^{-r_{0}}\right)}^{k}{\left(\log n\right)}^{r_{0}/4-1/4}\int_{\sqrt{\log n}}^{\infty }e^{-\pi t/2}dt\right)\\ \; & =O\left(z^{-r_{0}}{\left(p^{-r_{0}}+q^{-r_{0}}\right)}^{k}{\left(\log n\right)}^{r_{0}/4-1/4}e^{-\pi \sqrt{\log n}/2}\right)\\ \; & =O\left({\left(\log n\right)}^{r_{0}/4-1/4}e^{-\pi \sqrt{\log n}/2}\right), \end{array}$$
which is very small for large *n*. Note that for $t\in (\sqrt{\log n},\infty )$, $t^{r_{0}/2-1/2}$ is decreasing, and bounded above by ${\left(\log n\right)}^{r_{0}/4-1/4}$.
**Step two: Partitioning the integral**
There are now only finitely many saddle points to work with. We split the integral range into sub-intervals, each of which contains exactly one saddle point. This way, each integral has a contour traversing a single saddle point, and we will be able to estimate the dominant contribution in each integral from a small neighborhood around the saddle point. Assuming that $j^{*}$ is the largest *j* for which $\frac{2\pi j}{\log p/q}\leq \sqrt{\log n}$, we split the integral ${\tilde{E}}_{k}\left(z\right)$ as following
(116)$$\begin{array}{cc} {\tilde{E}}_{k}\left(z\right) & =-\frac{1}{2\pi }\left(\sum_{|j|<j^{*}}\int_{|t-t_{j}|\leq \frac{\pi }{\log p/q}}z^{-r_{0}+it}\Gamma \left(r_{0}+it\right){\left(p^{-r_{0}-it}+q^{-r_{0}-it}\right)}^{k}dt\right)\\ \; & -\frac{1}{2\pi }\int_{\frac{\pi }{\log p/q}\leq \left|t_{j}^{*}\right|<\sqrt{\log n}}\Gamma \left(r+it\right)z^{-r_{0}+it}{\left(p^{-r_{0}-it}+q^{-r_{0}-it}\right)}^{k}dt. \end{array}$$
By the same argument as in (Equation 115), the second term in (Equation 116) is also asymptotically negligible. Therefore, we are only left with
(117)$$\begin{array}{cc} {\tilde{E}}_{k}\left(z\right) & =\sum_{|j|<j^{*}}S_{j}\left(z\right), \end{array}$$
where $S_{j}\left(z\right)=-\frac{1}{2\pi }\int_{|t-t_{j}|\leq \frac{\pi }{\log p/q}}z^{-r_{0}+it}\Gamma \left(r_{0}+it\right){\left(p^{-r_{0}-it}+q^{-r_{0}-it}\right)}^{k}dt)$.
**Step three: Splitting the saddle contour**
For each integral $S_{j}$, we write the expansion of h(t) about $t_{j}$, as follows
(118)$$\begin{array}{cc} h(t) & =h\left(t_{j}\right)+\frac{1}{2}h^{{}^{\prime \prime }}\left(t_{j}\right){\left(t-t_{j}\right)}^{2}+O\left({\left(t-t_{j}\right)}^{3}\right). \end{array}$$
The main contribution for the integral estimate should come from an small integration path that reduces kh(t) to its quadratic expansion about $t_{j}$. In other words, we want the integration path to be such that
(119)$$\begin{array}{c} k{\left(t-t_{j}\right)}^{2}\to \infty ,\;\mathrm{and}\;k{\left(t-t_{j}\right)}^{3}\to 0. \end{array}$$
The above conditions are true when $|t-t_{j}|\gg k^{-1/2}$ and $\left|t-tj\right|\ll k^{-1/3}$. Thus, we choose the integration path to be $|t-t_{j}|\leq k^{-2/5}$. Therefore, we have
(120)$$\begin{array}{cc} S_{j}\left(z\right) & =-\frac{1}{2\pi }\int_{|t-t_{j}|\leq k^{-2/5}}z^{-r_{0}+it}\Gamma \left(r_{0}+it\right){\left(p^{-r_{0}-it}+q^{-r_{0}-it}\right)}^{k}dt\\ \; & \;\;\;-\frac{1}{2\pi }\int_{k^{-2/5}<\left|t-t_{j}\right|<\frac{\pi }{\log p/q}}z^{-r_{0}+it}\Gamma \left(r_{0}+it\right){\left(p^{-r_{0}-it}+q^{-r_{0}-it}\right)}^{k}dt. \end{array}$$
**Saddle Tails Pruning.**
We show that the integral is small for $k^{-2/5}<\left|t-t_{j}\right|<\frac{\pi }{\log p/q}$. We define
(121)$$\begin{array}{c} S_{j}^{\left(1\right)}\left(z\right)=-\frac{1}{2\pi }\int_{k^{-2/5}<\left|t-t_{j}\right|<\frac{\pi }{\log p/q}}z^{-r_{0}+it}\Gamma \left(r_{0}+it\right){\left(p^{-r_{0}-it}+q^{-r_{0}-it}\right)}^{k}dt. \end{array}$$
Note that for $|t-t_{j}|\leq \frac{\pi }{\log p/q}$, we have
(122)$$\begin{array}{cc} |p^{-r_{0}-it}+q^{-r_{0}-it}| & =\left(p^{-r_{0}}+q^{-r_{0}}\right)\;\;\sqrt{1-\frac{2p^{-r_{0}}q^{-r_{0}}}{{\left(p^{-r_{0}}+q^{-r_{0}}\right)}^{2}}(1-\cos\left(t\log p/q)\right)}\\ \; & \leq \left(p^{-r_{0}}+q^{-r_{0}}\right)\left(1-\frac{p^{-r_{0}}q^{-r_{0}}}{{\left(p^{-r_{0}}+q^{-r_{0}}\right)}^{2}}(1-\cos\left(t-t_{j})\log p/q\right)\right)\\ \; & \;\;\;\;\;\;\sin\mathrm{ce}\;\sqrt[{}]{1-x}\leq 1-\frac{x}{2}\;\mathrm{for}\;x\in [0,1]\\ \; & \leq \left(p^{-r_{0}}+q^{-r_{0}}\right)\left(1-\frac{2p^{-r_{0}}q^{-r_{0}}}{\pi^{2}{\left(p^{-r_{0}}+q^{-r_{0}}\right)}^{2}}{\left(\left(t-t_{j}\right)\log p/q\right)}^{2}\right)\\ \; & \;\;\;\;\;\sin\mathrm{ce}\;1-\cos x\geq \frac{2x^{2}}{\pi^{2}}\;\mathrm{for}\;|x|\leq \pi \\ \; & \leq \left(p^{-r_{0}}+q^{-r_{0}}\right)e^{-\gamma {\left(t-t_{j}\right)}^{2}}, \end{array}$$
where $\gamma =\frac{2p^{-r_{0}}q^{-r_{0}}\log^{2}p/q}{\pi^{2}{\left(p^{-r_{0}}+q^{-r_{0}}\right)}^{2}}$. Thus,
(123)$$\begin{array}{cc} S_{j}^{\left(1\right)}\left(z\right) & =O\left(z^{-r_{0}}|\Gamma \left(r_{0}+it\right)|\int_{k^{-2/5}<\left|t-t_{j}\right|<\frac{\pi }{\log p/q}}\left|p^{-r_{0}-it}+q^{-r_{0}-it}\right|dt\right)\\ \; & =O\left(z^{-r_{0}}{\left(p^{-r_{0}}+q^{-r_{0}}\right)}^{k}\int_{k^{-2/5}}^{\infty }e^{-\gamma ku^{2}}du\right)\\ \; & =O\left(z^{-r_{0}}{\left(p^{-r_{0}}+q^{-r_{0}}\right)}^{k}k^{-3/5}e^{-\gamma k^{1/5}}\right),\sin\mathrm{ce}\;\mathrm{erf}\left(x\right)=O\left(e^{-x^{2}}/x\right). \end{array}$$
**Central Approximation.**
Over the main path, the integrals are of the form
$$\begin{array}{cc} S_{j}^{\left(0\right)}\left(z\right) & =-\frac{1}{2\pi }\int_{|t-t_{j}|\leq k^{-2/5}}\Gamma \left(r_{0}+it\right)z^{-r_{0}+it}{\left(p^{-r_{0}-it}+q^{-r_{0}-it}\right)}^{k}dt\\ \; & =-\frac{1}{2\pi }\int_{|t-t_{j}|\leq k^{-2/5}}\Gamma \left(r_{0}+it\right)e^{-kh(t)}dt. \end{array}$$
We have
(124)$$\begin{array}{c} h^{{}^{\prime \prime }}\left(t_{j}\right)=\frac{\log^{2}p/q}{{\left({\left(p/q\right)}^{-r_{0}/2}+{\left(p/q\right)}^{r_{0}/2}\right)}^{2}}, \end{array}$$
and
(125)$$\begin{array}{c} p^{-r_{0}-it_{j}}+q^{-r_{0}-it_{j}}=p^{-it_{j}}\left(p^{-r_{0}}+q^{-r_{0}}\right). \end{array}$$
Therefore, by Laplace’s theorem (refer to [22]) we obtain
(126)$$\begin{array}{cc} S_{j}^{\left(0\right)}\left(z\right) & =\frac{1}{\sqrt{2\pi kh^{{}^{\prime \prime }}\left(t_{j}\right)}}\Gamma \left(r_{0}+it_{j}\right)e^{-kh(t_{j})}\left(1+O\left(k^{-1/2}\right)\right)\\ \; & =\frac{{\left(p/q\right)}^{-r_{0}/2}+{\left(p/q\right)}^{r_{0}/2}}{\sqrt[{}]{2\pi }\log p/q}\\ \; & \;\;\times z^{-r_{0}}{\left(p^{-r_{0}}+q^{-r_{0}}\right)}^{k}\Gamma \left(r_{0}+it_{j}\right)z^{-it_{j}}p^{-ikt_{j}}k^{-1/2}\left(1+O\left(\frac{1}{\sqrt{k}}\right)\right). \end{array}$$
We finally sum over all *j*$\left(|j|<j^{*}\right)$, and we get
(127)$$\begin{array}{cc} {\tilde{E}}_{k}\left(z\right) & =\frac{{\left(p/q\right)}^{-r_{0}/2}+{\left(p/q\right)}^{r_{0}/2}}{\sqrt[{}]{2\pi }\log p/q}\\ \; & \;\;\times \sum_{|j|<j^{*}}z^{-r_{0}}{\left(p^{-r_{0}}+q^{-r_{0}}\right)}^{k}\Gamma \left(r_{0}+it_{j}\right)z^{-it_{j}}p^{-ikt_{j}}k^{-1/2}\left(1+O\left(\frac{1}{\sqrt{k}}\right)\right). \end{array}$$
We can rewrite ${\tilde{E}}_{k}\left(z\right)$ as
(128)$$\begin{array}{cc} {\tilde{E}}_{k}\left(z\right) & =\Phi_{1}\left(\left(1+a\log p\right)\log_{p/q}n\right)\frac{z^{\nu }}{\sqrt{\log n}}\;\left(1+O\left(\frac{1}{\sqrt{\log n}}\right)\right), \end{array}$$
where $\nu =-r_{0}+a\log\left(p^{-r_{0}}+q^{-r_{0}}\right)$, and
(129)$$\begin{array}{cc} \Phi_{1}\left(x\right) & =\frac{{\left(p/q\right)}^{-r_{0}/2}+{\left(p/q\right)}^{r_{0}/2}}{\sqrt[{}]{2a\pi }\log p/q}\sum_{|j|<j^{*}}\Gamma \left(r_{0}+it_{j}\right)e^{-2\pi ijx}. \end{array}$$
For part ii, we move the line of integration to $r_{0}\in \left(0,\infty \right)$. Note that in this range, we must consider the contribution of the pole at s=0. We have
(130)$$\begin{array}{cc} {\tilde{E}}_{k}\left(z\right) & ={\mathrm{Res}}_{s=0}{\tilde{E}}_{k}^{*}\left(s\right)z^{-s}+\int_{r_{0}-i\infty }^{r_{0}+i\infty }{\tilde{E}}_{k}^{*}\left(z\right)z^{-s}ds. \end{array}$$
Computing the residue at s=0, and following the same analysis as in part *i* for the above integral, we arrive at
(131)$$\begin{array}{cc} {\tilde{E}}_{k}\left(z\right) & =2^{k}-\Phi_{1}\left(\left(1+a\log p\right)\log_{p/q}n\right)\frac{z^{\nu }}{\sqrt{\log n}}\;\left(1+O\left(\frac{1}{\sqrt{\log n}}\right)\right). \end{array}$$
For part iii. of Theorem 3, we shift the line of integration to $c_{0}\in \left(-2,-1\right)$, then we have
(132)$$\begin{array}{cc} {\tilde{E}}_{k}\left(z\right) & ={\mathrm{Res}}_{s=-1}{\tilde{E}}_{k}^{*}\left(s\right)z^{-s}+\int_{c-i\infty }^{c+i\infty }{\tilde{E}}_{k}^{*}\left(z\right)z^{-s}ds\\ \; & =z+O\left(z^{-c_{0}}{\left(p^{-c_{0}}+q^{-c_{0}}\right)}^{k}\right)\\ \; & =z^{a\log2}+O\left(z^{\nu_{0}}\right), \end{array}$$
where $\nu_{0}=-c_{0}+a\log\left(p^{-c_{0}}+q^{-c_{0}}\right)<1$.
**Step four: Asymptotic depoissonization**
To show that both conditions in (15) hold for ${\tilde{E}}_{k}\left(z\right)$, we extend the real values *z* to complex values $z=ne^{i\theta }$, where |θ|<π/2. To prove (Equation 103), we note that
(133)$$\begin{array}{c} |e^{-i\theta (r_{0}+it)}\Gamma \left(r_{0}+it\right){\left|=O(|t\right|}^{r_{0}-1/2}e^{t\theta -\pi |t|/2}), \end{array}$$
and therefore
(134)$$\begin{array}{cc} {\tilde{E}}_{k}\left(ne^{i\theta }\right) & =\frac{1}{2\pi }\int_{-\infty }^{\infty }e^{-i\theta (r_{0}+it)}n^{-r_{0}-it}\Gamma \left(r_{0}+it\right){\left(p^{-r_{0}-it}+q^{-r_{0}-it}\right)}^{k}dt \end{array}$$
is absolutely convergent for |θ|<π/2. The same saddle point analysis applies here and we obtain
(135)$$\begin{array}{cc} |{\tilde{E}}_{k}\left(z\right)| & \leq B\frac{|z^{\nu }|}{\sqrt{\log n}}, \end{array}$$
where $B=|\Phi_{1}\left(\left(1+a\log p\right)\log_{p/q}n\right)|$, and ν is as in (Equation 128). Condition (Equation 103) is therefore satisfied. To prove condition (Equation 104) We see that for a fixed *k*,
(136)$$\begin{array}{c} |{\tilde{E}}_{k}\left(z\right)e^{z}|\leq \sum_{w\in A^{k}}\left|e^{z}-e^{z(1-P(w))}\right|\\ \leq 2^{k+1}e^{\left|z|\cos(\theta \right)}. \end{array}$$
Therefore, we have
(137)$$\begin{array}{c} E\left[{\hat{X}}_{n,k}\right]=\tilde{E}\left(n\right)+O\left(\frac{n^{\nu -1}}{\sqrt{\log n}}\right). \end{array}$$
This completes the proof of Theorem 3. □

**On the Second Factorial Moment:** We poissonize the sequence ${\left(E\left[{\left({\hat{X}}_{n,k}\right)}_{2}\right]\right)}_{n\geq 0}$ as well. By the analysis in (Equation 27),
$$\begin{array}{cc} E[{\left({\hat{X}}_{n,k}\right)}_{2}] & =\sum_{\begin{array}{c} w,w^{\prime }\in A^{k}\\ w\neq w^{\prime } \end{array}}\left(1-{\left(1-P\left(w\right)\right)}^{n}-{\left(1-P\left(w^{\prime }\right)\right)}^{n}+{\left(1-P\left(w\right)-P\left(w^{\prime }\right)\right)}^{n}\right), \end{array}$$
which gives the following poissonized form
(138)$$\begin{array}{cc} \tilde{G}\left(z\right) & =\sum_{n\geq 0}E\left[{\left({\hat{X}}_{n,k}\right)}_{2}\right]\frac{z^{n}}{n!}e^{-z}\\ \; & =\sum_{\begin{array}{c} w,w^{\prime }\in A^{k}\\ w\neq w^{\prime } \end{array}}1-e^{-P(w)z}-e^{-P(w^{\prime })z}+e^{-(P\left(w\right)+P\left(w^{\prime }\right))z}\\ \; & =\sum_{\begin{array}{c} w,w^{\prime }\in A^{k}\\ w\neq w^{\prime } \end{array}}\left(1-e^{-P(w^{\prime })z}\right)\left(1-e^{-P(w)z}\right)\\ \; & ={\left(\sum_{w\in A^{k}}\left(1-e^{-P(w)z}\right)\right)}^{2}-\sum_{w\in A^{k}}{\left(1-e^{-P(w)z}\right)}^{2}\\ \; & ={\left({\tilde{E}}_{k}\left(z\right)\right)}^{2}-\sum_{w\in A^{k}}{\left(1-e^{-P(w)z}\right)}^{2}\\ \; & ={\left({\tilde{E}}_{k}\left(z\right)\right)}^{2}-\sum_{w\in A^{k}}\left(1-2e^{-P(w)z}+e^{-2P(w)z}\right). \end{array}$$
We show that in all ranges of *a* the leftover sum in (Equation 138) has a lower order contribution to ${\tilde{G}}_{k}\left(z\right)$ compared to ${\left({\tilde{E}}_{k}\left(z\right)\right)}^{2}$. We define
(139)$$\begin{array}{c} {\tilde{L}}_{k}\left(z\right)=\sum_{w\in A^{k}}\left(1-2e^{-P(w)z}+e^{-2P(w)z}\right). \end{array}$$
In the first range for *k*, we take the Mellin transform of ${\tilde{L}}_{k}\left(z\right)$, which is
(140)$$\begin{array}{cc} {\tilde{L}}_{k}^{*}\left(s\right) & =-2\Gamma \left(s\right)\sum_{w\in A^{k}}P{\left(w\right)}^{-s}+\Gamma \left(s\right)\sum_{w\in A^{k}}{\left(2P\left(w\right)\right)}^{-s}\\ \; & =-2\Gamma \left(s\right){\left(p^{-s}+q^{-s}\right)}^{k}+\Gamma \left(s\right)2^{-s}{\left(p^{-s}+q^{-s}\right)}^{k}\\ \; & =\Gamma \left(s\right){\left(p^{-s}+q^{-s}\right)}^{k}\left(2^{-s-1}-1\right), \end{array}$$
and we note that the fundamental strip for this Mellin transform of is 〈−2,0〉 as well. The inverse Mellin transform for c∈(−2,0) is
(141)$$\begin{array}{cc} {\tilde{L}}_{k}\left(z\right) & =\frac{1}{2\pi i}\int_{c-i\infty }^{c+i\infty }{\tilde{L}}_{k}^{*}\left(s\right)z^{-s}ds\\ \; & =\frac{1}{\pi i}\int_{c-i\infty }^{c+i\infty }\Gamma \left(s\right){\left(p^{-s}+q^{-s}\right)}^{k}\left(2^{-s-1}-1\right)z^{-s}ds \end{array}$$
We note that this range of $r_{0}$ corresponds to
(142)$$\begin{array}{c} \frac{2}{\log q^{-1}+\log p^{-1}}<a<\frac{p^{2}+q^{2}}{q^{2}\log q^{-1}+p^{2}\log p^{-1}}. \end{array}$$
The integrand in (Equation 141) is quite similar to the one seen in (Equation 107). The only difference is the extra term $2^{-s-1}-1$. However, we notice that $2^{-s-1}-1$ is analytic and bounded. Thus, we obtain the same saddle points with the real part as in (Equation 109) and the same imaginary parts in the form of $\frac{2\pi ij}{\log p/q}$, j∈Z. Thus, the same saddle point analysis for the integral in (Equation 107) applies to ${\tilde{L}}_{k}\left(z\right)$ as well. We avoid repeating the similar steps, and we skip to the central approximation, where by Laplace’s theorem (ref. [22]), we get
(143)$$\begin{array}{cc} {\tilde{L}}_{k}\left(z\right) & =\frac{{\left(p/q\right)}^{-r_{0}/2}+{\left(p/q\right)}^{r_{0}/2}}{\sqrt[{}]{2\pi }\log p/q}\\ \; & \;\times \sum_{|j|<j^{*}}z^{-r_{0}}{\left(p^{-r_{0}}+q^{-r_{0}}\right)}^{k}\left(2^{-r_{0}-1-it_{j}}-1\right)\\ \; & \;\;\;\times \Gamma \left(r_{0}+it_{j}\right)z^{-it_{j}}p^{-ikt_{j}}k^{-1/2}\left(1+O\left(\frac{1}{\sqrt{k}}\right)\right), \end{array}$$
which can be represented as
(144)$$\begin{array}{cc} {\tilde{L}}_{k}\left(z\right) & =\Phi_{2}\left(\left(1+a\log p\right)\log_{p/q}n\right)\frac{z^{\nu }}{\sqrt{\log n}}\;\left(1+O\left(\frac{1}{\sqrt{\log n}}\right)\right), \end{array}$$
where
(145)$$\begin{array}{cc} \Phi_{2}\left(x\right) & =\frac{{\left(p/q\right)}^{-r_{0}/2}+{\left(p/q\right)}^{r_{0}/2}}{\sqrt[{}]{2a\pi }\log p/q}\sum_{|j|<j^{*}}\left(2^{-r_{0}-1-it_{j}}-1\right)\Gamma \left(r_{0}+it_{j}\right)e^{-2\pi ijx}. \end{array}$$
This shows that ${\tilde{L}}_{k}\left(z\right)=O\left(\frac{z^{\nu }}{\sqrt{\log n}}\right)$, when
$$\frac{2}{\log q^{-1}+\log p^{-1}}<a<\frac{p^{2}+q^{2}}{q^{2}\log q^{-1}+p^{2}\log p^{-1}}.$$
Subsequently, for $\frac{1}{\log q^{-1}}<a<\frac{2}{\log q^{-1}+\log p^{-1}}$, we get
(146)$$\begin{array}{cc} {\tilde{L}}_{k}\left(z\right) & =2^{k}-\Phi_{2}\left(\left(1+a\log p\right)\log_{p/q}n\right)\frac{z^{\nu }}{\sqrt{\log n}}\;\left(1+O\left(\frac{1}{\sqrt{\log n}}\right)\right), \end{array}$$
and for $\frac{p^{2}+q^{2}}{q^{2}\log q^{-1}+p^{2}\log p^{-1}}<a<\frac{1}{\log p^{-1}}$, we get
(147)$$\begin{array}{cc} {\tilde{L}}_{k}\left(z\right) & =O(n^{2}). \end{array}$$
It is not difficult to see that for each range of *a* as stated above, ${\tilde{L}}_{k}\left(z\right)$ has a lower order contribution to the asymptotic expansion of ${\tilde{G}}_{k}\left(z\right)$, compared to ${\left({\tilde{E}}_{k}\left(z\right)\right)}^{2}$. Therefore, this leads us to Theorem 4, which will be proved bellow.

**Proof of Theorem 4.** It is only left to show that the two depoissonization conditions hold: For condition (Equation 103) in Theorem 15, from (Equation 135) we have
(148)$$\begin{array}{cc} |{\tilde{G}}_{k}\left(z\right)| & \leq B^{2}\frac{|z^{2\nu }|}{\log n}, \end{array}$$
and for condition (Equation 104), we have, for fixed *k*,
(149)$$\begin{array}{cc} |{\tilde{G}}_{k}\left(z\right)e^{z}| & \leq \sum_{\begin{array}{c} w,w^{\prime }\in A^{k}\\ w\neq w^{\prime } \end{array}}\left|e^{z}-e^{(1-P(w))z}-e^{(1-P\left(w^{\prime }\right))z}+e^{(1-\left(P\left(w\right)+P\left(w^{\prime }\right)\right))z}\right|\\ \; & \leq 4^{k}e^{|z|\cos\theta }. \end{array}$$
Therefore both depoissonization conditions are satisfied and the desired result follows. □


**Corollary. A Remark on the Second Moment and the Variance**


For the second moment we have
(150)$$\begin{array}{cc} E\left[{\left({\hat{X}}_{n,k}\right)}^{2}\right] & =\sum_{\begin{array}{c} w,w^{\prime }\in A^{k}\\ w\neq w^{\prime } \end{array}}E\left[{\hat{X}}_{n,k}^{\left(w\right)}{\hat{X}}_{n,k}^{\left(w^{\prime }\right)}\right]+\sum_{w\in A^{k}}E\left[{\hat{X}}_{n,k}^{\left(w\right)}\right]\\ \; & =\sum_{\begin{array}{c} w,w^{\prime }\in A^{k}\\ w\neq w^{\prime } \end{array}}\left(1-{\left(1-P\left(w\right)\right)}^{n}-{\left(1-P\left(w^{\prime }\right)\right)}^{n}+{\left(1-P\left(w\right)-P\left(w^{\prime }\right)\right)}^{n}\right)\\ \; & \;\;+\sum_{w\in A^{k}}\left(1-{\left(1-P(w)\right)}^{n}\right). \end{array}$$
Therefore, by (Equation 105) and (Equation 138) the Poisson transform of the second moment, which we denote by ${\tilde{G}}_{k}^{\left(2\right)}\left(z\right)$ is
(151)$$\begin{array}{cc} {\tilde{G}}_{k}^{\left(2\right)}\left(z\right) & ={\left({\tilde{E}}_{k}\left(z\right)\right)}^{2}+{\tilde{E}}_{k}\left(z\right)-\sum_{w\in A^{k}}\left(1-2e^{-P(w)z}+e^{-2P(w)z}\right), \end{array}$$
which results in the same first order asymptotic as the second factorial moment. Also, it is not difficult to extend the proof in Chapter 6 to show that the second moments of the two models are asymptotically the same. For the variance we have
(152)$$\begin{array}{cc} \mathrm{Var}[{\hat{X}}_{n,k}] & =E\left[{\left({\hat{X}}_{n,k}\right)}^{2}\right]-{\left(E\left[{\hat{X}}_{n,k}\right]\right)}^{2}\\ \; & =\sum_{\begin{array}{c} w,w^{\prime }\in A^{k}\\ w\neq w^{\prime } \end{array}}\left(1-{\left(1-P\left(w\right)\right)}^{n}-{\left(1-P\left(w^{\prime }\right)\right)}^{n}+{\left(1-P\left(w\right)-P\left(w^{\prime }\right)\right)}^{n}\right)\\ \; & \;\;+\sum_{w\in A^{k}}\left(1-{\left(1-P(w)\right)}^{n}\right)\\ \; & -\sum_{\begin{array}{c} w,w^{\prime }\in A^{k}\\ w\neq w^{\prime } \end{array}}\left(1-{\left(1-P\left(w\right)\right)}^{n}-{\left(1-P\left(w^{\prime }\right)\right)}^{n}+{\left(1-P\left(w\right)-P\left(w^{\prime }\right)\right)}^{n}\right)\\ \; & \;\;-\sum_{w\in A^{k}}\left(1-{\left(1-P(w)\right)}^{n}-{\left(1-P(w)\right)}^{n}+{\left(1-P(w)\right)}^{2n}\right)\\ \; & =\sum_{w\in A^{k}}\left({\left(1-P(w)\right)}^{n}-{\left(1-P(w)\right)}^{2n}\right). \end{array}$$
Therefore the Poisson transform, which we denote by ${\tilde{G}}_{k}^{\mathrm{var}}\left(z\right)$ is
(153)$$\begin{array}{c} {\tilde{G}}_{k}^{\mathrm{var}}\left(z\right)=\sum_{w\in A^{k}}\left(e^{-P(w)z}-e^{-(2P\left(w\right)+{\left(P\left(w\right)\right)}^{2})z}\right). \end{array}$$
The Mellin transform of the above function has the following form
(154)$$\begin{array}{cc} {\tilde{G^{*}}}_{k}^{\mathrm{var}}\left(z\right) & =\Gamma \left(s\right){\left(p^{-s}+q^{-s}\right)}^{k}\left(-1+O\left(P\left(w\right)\right)\right). \end{array}$$
This is quite similar to what we saw in (Equation 106), which indicates that the variance has the same asymptotic growth as the expected value. But the variance of the two models do not behave in the same way (cf. Figure 2).

## 4. Summary and Conclusions

We studied the first-order asymptotic growth of the first two (factorial) moments of the *k*th Subword Complexity. We recall that the *k*th Subword Complexity of a string of length *n* is denoted by $X_{n,k}$, and is defined as the number of distinct subwords of length *k*, that appear in the string. We are interested in the asymptotic analysis for when *k* grows as a function of the string’s length. More specifically, we conduct the analysis for k=Θ(logn), and as n→∞.

The analysis is inspired by the earlier work of Jacquet and Szpankowski on the analysis of suffix trees, where they are compared to independent tries (cf. [14]). In our work, we compare the first two moments of the *k*th Subword Complexity to the *k*th Prefix Complexity over a random trie built over *n* independently generated binary strings. We recall that we define the *k*th Prefix Complexity as the number of distinct prefixes that appear in the trie at level *k* and lower.

We obtain the generating functions representing the expected value and the second factorial moments as their coefficients, in both settings. We prove that the first two moments have the same asymptotic growth in both models. For deriving the asymptotic behavior, we split the range for *k* into three intervals. We analyze each range using the saddle point method, in combination with residue analysis. We close our work with some remarks regarding the comparison of the second moment and the variance to the *k*th Prefix Complexity.

## 5. Future Challenges

The intervals’ endpoints for *a* in Theorems 3 and 4 are not investigated in this work. The asymptotic analysis of the end points can be studied using van der Waerden saddle point method [24].

The analogous results are not (yet) known in the case where the underlying probability source has Markovian dependence or in the case of dynamical sources. 

## Figures and Tables

**Figure 1 entropy-22-00207-f001:**
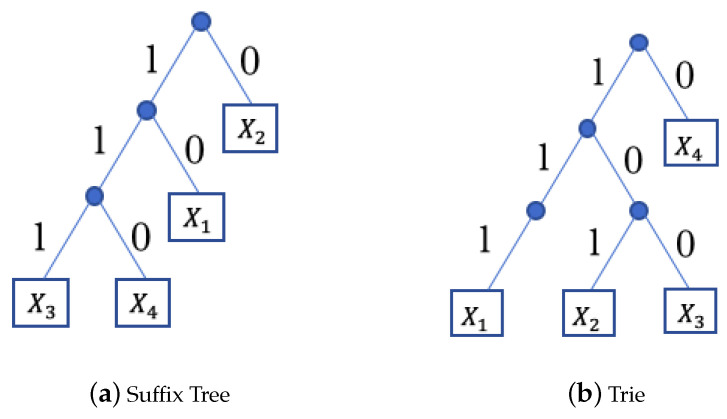
The suffix tree in (**a**) is built over the first four suffixes of string X=101110..., and the trie in (**b**) is build over strings $X_{1}=111...$, $X_{2}=101...$, $X_{3}=100$, and $X_{4}=010...$.

**Figure 2 entropy-22-00207-f002:**
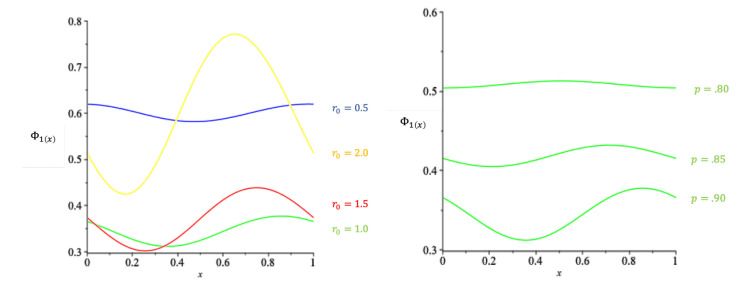
**Left**: $\Phi_{1}\left(x\right)$ at p=0.90, and various levels of $r_{0}$. The amplitude increases as $r_{0}$ increases. **Right**: $\Phi_{1}\left(x\right)$ at $r_{0}=1$, and various levels of *p*. The amplitude tends to zero as $p\to 1/2^{+}$.

**Figure 3 entropy-22-00207-f003:**
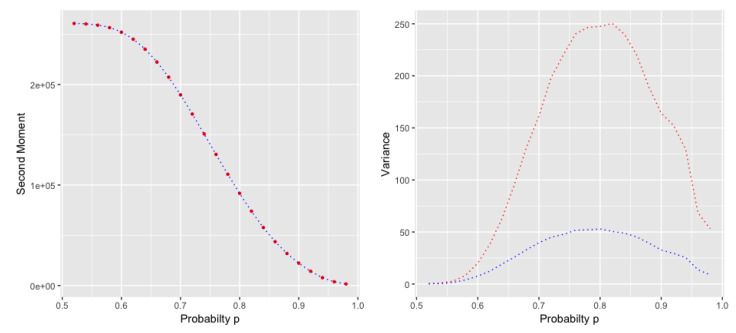
Approximated second moments (**left**), and variances (**right**) of the *k*th Subword Complexity (**red**), and the *k*th Prefix Complexity (**blue**), for n=4000, at different probability levels, averaged over 10,000 iterations.

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
