# Peer review of "Asymptotic Analysis of the kth Subword Complexity"

_entropy, 2020, doi:10.3390/e22020207_

Round 1
Reviewer 1 Report
I would like to thank the authors for addressing the issues mentioned in the previous review.
In order to make the paper more accessible to readers that are not highly familiar with the topic, the paper should define the concept of "memory-less source". The paper should also define concepts such as "first and second factorial moments".
At line 42, the papers mentions dividing the analysis in three distinct intervals, based on the value of k. A reason for taking this path should be provided. It can also be observed that the cases in which k is equal to the ends of the specified intervals are not considered. A reason should be included for this choice.
A clear example proving how the results included in the paper can be applied in practice should be included. The paper mentions between the lines 28 and 34, why analyzing sub-word complexity is useful, but no concrete example is given in relation to the proposed approach.
Author Response
We thank the reviewer for their careful reading of the manuscript and their constructive remarks. We have taken the comments on board to improve and clarify the manuscript. Please find below our response to all comments:
1. In order to make the paper more accessible to readers that are not highly familiar with the topic, the paper should define the concept of "memory-less source".
Please refer to lines 102,103.
2.The paper should also define concepts such as "first and second factorial moments".
This issue is addressed in lines 105-108.
3. At line 42, the papers mention dividing the analysis in three distinct intervals, based on the value of k. A reason for taking this path should be provided.
Lines 115-122 clarify that the rate of growth changes depending on the intervals. A more in depth explanation can be found in the proof of Theorem 3 on page 25.
4. It can also be observed that the cases in which k is equal to the ends of the specified intervals are not considered. A reason should be included for this choice.
This is added in future challenges, lines 243, 244.
5.A clear example proving how the results included in the paper can be applied in practice should be included. The paper mentions between the lines 28 and 34, why analyzing sub-word complexity is useful, but no concrete example is given in relation to the proposed approach.
Please refer to lines 35-39, along with references 11, 12.
Reviewer 2 Report
All issues have been addressed. I have no more suggestions.
Author Response
We appreciate the positive feedback from the reviewer.
Round 2
Reviewer 1 Report
I would like to thank the authors for thoroughly addressing the comments in the previous review.This manuscript is a resubmission of an earlier submission. The following is a list of the peer review reports and author responses from that submission.
Round 1
Reviewer 1 Report
This is an interesting work on subword complexity. However, the paper
presents a lot of equations with little insights on the importance of the topic being approached. In order to make this article more accessible to a wider audience, I suggest the authors to include additional information on why it is important to compute subword complexity and which are possible practical approaches. I see that subword complexity could be used e.g. to analyze text complexity. The authors should mention related works that could benefit from their approach,e.g.: doi: 10.1209/0295-5075/100/58002 and 10.3233/JAD-150520. In addition, I miss a better motivation for this work. Finally, the authors should also check the text
Author Response
The authors sincerely appreciate the feedback from the editors and referees, and we have made a concerted effort to take all suggested changes into account:
The introduction is revised to address:
the important to compute subword complexity,
related works that could benefit from our approach,
and to provide better motivation for this work.
Reviewer 2 Report
The abstract of the paper should be extended in order to better convey its results to users less familiar with the topic.
At line 37, the authors include a long series of references, without including any details regarding them. At least a few words should be added for each reference for guiding the reader.
The literature review is virtually not existent and does not adequately discuss the gaps in the scientific literature and how the authors have tried to address them. A discussion should be added in order to better justify the need for this research.
The entire structure of the paper should be revised in order to follow the "classic" structure of a research paper. After the introduction and literature review, the methods should be described in direct connection with the literature, followed by simulations and results discussion and eventually some practical applications. The paper is also ending "abruptly" with no Conclusions section. If possible, the authors are kindly asked to also include possible future research directions.
The purpose of variable a is not clearly stated in the results section. Please carefully state the purpose of all the introduced variables.
Author Response
The authors sincerely appreciate the feedback from the editors and referees, and we have made a concerted effort to take all suggested changes into account:
Line 37 is removed and the introduction section is edited to address details regarding references.
Lines 51-80 added to address the issue regarding the literature review.
The structure of the paper is revised to follow the "classic" structure of a research paper.
Future research directions is now added to the manuscript.